# Importance Sampling for Nonlinear Models

**Prakash P. Rajmohan** [1]  **Fred Roosta** [2 3]

## Abstract

While norm-based and leverage-score-based methods have been extensively studied for identifying "important" data points in linear models, analogous tools for nonlinear models remain significantly underdeveloped. By introducing the concept of the adjoint operator of a nonlinear map, we address this gap and generalize norm-based and leverage-score-based importance sampling to nonlinear settings. We demonstrate that sampling based on these generalized notions of norm and leverage scores provides approximation guarantees for the underlying nonlinear mapping, similar to linear subspace embeddings. As direct applications, these nonlinear scores not only reduce the computational complexity of training nonlinear models by enabling efficient sampling over large datasets but also offer a novel mechanism for model explainability and outlier detection. Our contributions are supported by both theoretical analyses and experimental results across a variety of supervised learning scenarios.

## 1. Introduction

The process of training in machine learning (ML) typically boils down to solving an optimization problem of the form

$$\min_{\boldsymbol{\theta} \in \mathbb{R}^p} \left\{ \mathcal{L}(\boldsymbol{\theta}) = \sum_{i=1}^{n} \ell(f_i(\boldsymbol{\theta})) \right\}, \tag{1}$$

where $\ell$ is a loss function, and $f_i : \mathbb{R}^p \to \mathbb{R}$ is a potentially nonlinear mapping, parametrized by $\boldsymbol{\theta}$, that represents the ML model evaluated at the $i^{\text{th}}$ training data point. Through-

out this paper, we make the umbrella assumption that $n \geq p$, i.e., we operate in the underparameterized setting.

For example, in the simple linear regression, $f_i(\boldsymbol{\theta}) = \langle \boldsymbol{\theta}, \mathbf{x}_i \rangle - y_i$, where $(\mathbf{x}_i, y_i)$ is the $i^{\text{th}}$ input-output pair. Using the squared loss $\ell(t) = t^2$ gives rise to the familiar linear least-squares problem $\mathcal{L}(\boldsymbol{\theta}) = \|\mathbf{X}\boldsymbol{\theta} - \mathbf{y}\|^2$, where $\mathbf{X} \in \mathbb{R}^{n \times d}$ is the input data matrix whose $i^{\text{th}}$ row is $\mathbf{x}_i$, and $\mathbf{y} \in \mathbb{R}^n$ is the output vector whose $i^{\text{th}}$ component is $y_i$.

The growth of ML over the past decade has been largely driven by the explosion in the availability of data. However, this exponential increase in data volume presents significant computational challenges, particularly in solving (1) and developing diagnostic tools for post-training analysis. These challenges have been extensively studied and addressed in simpler linear settings. In this context, randomized numerical linear algebra (RandNLA) has emerged as a powerful paradigm for approximating underlying matrices and accelerating computations. RandNLA has led to significant advancements in algorithms for fundamental matrix problems, including matrix multiplication, least-squares problems, least-absolute deviation, and low-rank matrix approximation, among others. For further reading, see the lecture notes and surveys on this topic, such as Mahoney et al. (2011); Woodruff et al. (2014); Drineas & Mahoney (2018); Martinsson & Tropp (2020); Murray et al. (2023); Dereziński & Mahoney (2024).

Arguably, linear least-squares problems have been among the most well-studied problems in RandNLA (Sarlos, 2006; Nelson & Nguyên, 2013; Meng & Mahoney, 2013; Clarkson & Woodruff, 2017). Various randomized approximation techniques, ranging from data-independent oblivious methods (e.g., sketching and projections) to data-dependent non-oblivious sampling methods (e.g., non-uniform leverage score or row-norm sampling), have emerged as powerful tools to speed up computations directly (Woodruff et al., 2014) or to construct preconditioners for downstream linear algebra subroutines (Avron et al., 2010). While these tools have proven remarkably successful in accelerating computations for linear least-squares problems, extending them to more general problems of the form (1) remains challenging. Efforts to push these boundaries into the nonlinear realm often remain ad hoc and limited in scope (Gajjar & Musco, 2021; Erdélyi et al., 2020; Avron et al., 2019; 2017). A key

---
[1]School of Electrical Engineering and Computer Science, University of Queensland, Brisbane, Australia. [2]School of Mathematics and Physics, University of Queensland, Brisbane, Australia. [3]ARC Training Centre for Information Resilience (CIRES), Brisbane, Australia. Correspondence to: Prakash P. Rajmohan <p.palanivelurajmohan@uq.net.au>, Fred Roosta <fred.roosta@uq.edu.au>.

*Proceedings of the $42^{nd}$ International Conference on Machine Learning*, Vancouver, Canada. PMLR 267, 2025. Copyright 2025 by the author(s).

gap lies in the lack of a systematic framework to capture and embed the "nonlinear component" of the objective function while preserving the critical approximation properties well-established in linear embeddings.

In this paper, we aim to bridge this gap to some extent. Specifically, we focus on non-uniform sampling in (1). Letting $\boldsymbol{\theta}^\star$ and $\boldsymbol{\theta}_{\mathcal{S}}^\star$ denote the optimal parameters obtained from training the model over the full dataset and a, potentially non-uniformly, sampled subset of data, respectively, our goal is to ensure that, for any small $\varepsilon$, the samples are selected such that

$$\mathcal{L}(\boldsymbol{\theta}_{\mathcal{S}}^\star) \leq \mathcal{L}(\boldsymbol{\theta}^\star) + \mathcal{O}\left(\varepsilon\right). \tag{2}$$

Recent work by (Gajjar et al., 2023; 2024) has made progress toward this goal by proposing a one-shot active learning strategy using leverage scores of the data for certain classes of single-neuron predictors. Here, we take a step further by introducing the concept of the adjoint operator of a nonlinear map. This allows us to generalize norm-based and leverage-score-based importance sampling to nonlinear settings, thereby obtaining approximation guarantees of the form (2) in many settings.

**Contributions**. Our contributions are as follows:

1. By introducing the nonlinear adjoint operator, we provide a systematic framework for constructing importance sampling scores for (1). While numerical integration can generally approximate these scores, we show that for specific nonlinear models, they can be derived directly.

2. We further show that, under certain assumptions, importance sampling based on these scores achieves (2). To our knowledge, this is the first work to extend such guarantees to NNs.

3. To validate the theoretical results, we present experiments for several supervised learning tasks. We show that, beyond reducing computational costs, our framework can be used post hoc for diagnostics, such as identifying important samples and detecting anomalies.

**Notation.** Vectors and matrices are denoted by bold lowercase and bold uppercase letters, respectively. The $i^{th}$ row of a data matrix $\mathbf{X} \in \mathbb{R}^{n \times d}$ is denoted by $\mathbf{x}_i \in \mathbb{R}^d$. The pseudoinverse of $\mathbf{X}$ is denoted by $\mathbf{X}^\dagger$, and the $i^{\text{th}}$ standard basis vector in $\mathbb{R}^n$ is denoted by $\mathbf{e}_i$. The spectral norm and Frobenius norm of a matrix are denoted by $\|\cdot\|_2$ and $\|\cdot\|_{\mathrm{F}}$, respectively. The inner product between two vectors is denoted by $\langle\cdot,\cdot\rangle$. We use $\mathcal{O}(\cdot)$ for Big-O complexity and $\widetilde{\mathcal{O}}(\cdot)$ to omit logarithmic factors. The sub-sampled data matrix is represented by $\mathbf{X}_{\mathcal{S}} \in \mathbb{R}^{s \times d}$, where $s$ is the number of sub-sampled points.

## 2. Background and Related Work

**Importance Sampling in Linear Models.** In linear settings, importance sampling has been widely used for various linear algebra tasks such as matrix multiplication (Drineas et al., 2006a), least-squares regression (Drineas et al., 2006b), and low-rank approximation (Cohen et al., 2017). It approximates large data matrices by prioritizing data points with high "information", reducing computational (Clarkson & Woodruff, 2017; Nelson & Nguyên, 2013) and storage costs (Iwen et al., 2021; Meng & Mahoney, 2013) while ensuring strong guarantees for downstream tasks.

Formally, let $\mathbf{X} \in \mathbb{R}^{n \times d}$, $n \gg d$, be a data matrix with row vectors $\mathbf{x}_i \in \mathbb{R}^d$. Each row is assigned a nonnegative weight $q_i \geq 0$ through an importance sampling scheme, with sampling probability $\tau_i = q_i / \sum_{j=1}^n q_j$. Consider selecting $s \ll n$ rows of $\mathbf{X}$ independently at random, with replacement, according to $\{\tau_i\}$. Let $\mathbf{X}_{\mathcal{S}} \in \mathbb{R}^{s \times d}$ be the sub-sampled matrix, where the $i^{\text{th}}$ row is $\mathbf{x}_i / \sqrt{s\,\tau_i}$. If $\{\tau_i\}$ is constructed appropriately, $\mathbf{X}_{\mathcal{S}}$ preserves key spectral and geometric properties of $\mathbf{X}$ with high probability, acting as a lower-dimensional approximation. This is captured by a *linear subspace embedding* guarantee: for all $\mathbf{v} \in \mathbb{R}^d$,

$$(1 - \varepsilon)\|\mathbf{X}\mathbf{v}\|_2^2 \leq \|\mathbf{X}_{\mathcal{S}}\mathbf{v}\|_2^2 \leq (1 + \varepsilon)\|\mathbf{X}\mathbf{v}\|_2^2. \tag{3}$$

Two common approaches for constructing $\{\tau_i\}$ are based on the *norms* and *leverage scores* of the rows of $\mathbf{X}$. In row-norm sampling, we set $q_i = \|\mathbf{x}_i\|_2^2$, meaning rows that contribute more to the overall $\ell_2$-energy, $\|\mathbf{X}\|_{\mathrm{F}}^2 = \sum_{i=1}^n \|\mathbf{x}_i\|_2^2$, are sampled more frequently. Leverage scores quantify the influence of each row $\mathbf{x}_i$ on the row space of $\mathbf{X}$ and are defined as

$$q_i = \left\langle \mathbf{e}_i, \mathbf{X}\mathbf{X}^\dagger \mathbf{e}_i \right\rangle = \min_{\boldsymbol{\alpha} \in \mathbb{R}^n} \|\boldsymbol{\alpha}\|_2^2 \text{ subject to } \mathbf{X}^\mathsf{T}\boldsymbol{\alpha} = \mathbf{x}_i.$$

Thus, the $i^{\text{th}}$ leverage score captures the "importance" of $\mathbf{x}_i$ in spanning the data subspace.

These importance sampling schemes are particularly effective when the data exhibits high coherence (Paschou et al., 2007; Mahoney & Drineas, 2009; Gittens & Mahoney, 2013; Fan et al., 2011; Eshragh et al., 2022). It has been well established that, for any $\varepsilon \in (0, 1)$, as long as the sample size $s$ is sufficiently large–specifically, $s \in \mathcal{O}\left(d \ln d / \varepsilon^2\right)$–the approximation in (3) holds with high probability (Martinsson & Tropp, 2020). These bounds are tight (up to logarithmic factors) (Woodruff et al., 2014; Chen & Price, 2019).

The linear subspace embedding guarantee in (3) forms the foundation for approximating linear least-squares problems (Woodruff et al., 2014). However, for loss functions beyond least-squares, simple linear subspace embedding may not provide the desired guarantees. In such cases, alternative tools like importance sampling scores (e.g., Lewis weights (Apers et al., 2024; Johnson & Schechtman, 2001; Bourgain

et al., 1989; Cohen & Peng, 2015)) or coresets (Feldman, 2020; Mirzasoleiman et al., 2020; Lucic et al., 2018; Har-Peled & Mazumdar, 2004) are commonly used.

In particular for coresets, Langberg & Schulman (2010) introduced a sensitivity sampling framework that provides foundational coreset guarantees for broad classes of objectives. This approach (akin to importance sampling) assigns each data point a sampling probability proportional to its worst-case influence (sensitivity) on the objective function, yielding a coreset that achieves a $(1 \pm \varepsilon)$ approximation for all queries. Subsequent work has refined this idea for specific loss functions; for instance, Tremblay et al. (2019) combine sensitivity scores with determinantal point processes (DPP) to reduce redundant draws and thus promote diversity among the selected points. For logistic regression, Munteanu et al. (2018) prove that although no sublinear-size coreset exists in the worst case, a sensitivity-based scheme can yield the first provably sublinear approximation coreset when the dataset's complexity measure $\mu(X)$ is bounded. These results, position importance-sampling coresets as the natural extension of linear embeddings when moving beyond least-squares.

For more general loss functions $\ell$, some works have extended approximation guarantees similar to (2) to specific problems, such as logistic regression (Samadian et al., 2020; Huggins et al., 2016; Tolochinksy et al., 2022), linear predictor models with hinge-like loss (Mai et al., 2021), $\ell_p$ regression (Chen & Derezinski, 2021; Musco et al., 2022), and kernel regression (Erdélyi et al., 2020).

**Importance Sampling in Nonlinear Models.** Importance sampling techniques have long played a crucial role in modern large-scale machine learning tasks, serving as tools to identify the most important samples and accelerating optimization procedures (Katharopoulos & Fleuret, 2018; Stich et al., 2017; Nabian et al., 2021; Canévet et al., 2016; Liu & Lee, 2017; Meng et al., 2022; Xu et al., 2016; Liu et al., 2024). However, their use in obtaining approximation guarantees of the form (2) has been very limited.

To our knowledge, Gajjar et al. (2023; 2024) are among the first to consider non-linear function families and analyze a leverage-score-based sampling scheme (also referred to as a one-shot active learning sampling scheme) in the context of single-index (or "single neuron") models, where $\ell$ is the squared loss and $f_i(\boldsymbol{\theta}) = \phi(\langle \boldsymbol{\theta}, \mathbf{x}_i \rangle)$ for some scalar non-linearity $\phi \colon \mathbb{R} \to \mathbb{R}$. For their derivations, the authors employ intricate tools from high-dimensional probability, as the mapping $\mathbf{x} \mapsto \phi(\langle \mathbf{w}, \mathbf{x} \rangle)$ introduces non-linearity that invalidates the straightforward matrix Chernoff arguments typically used in linear settings.

Gajjar et al. (2023) showed that if $\phi$ is $L$-Lipschitz, one can collect $\widetilde{\mathcal{O}}(d^2/\varepsilon^4)$ labels, sampled with probabilities

proportional to the linear leverage scores of $\mathbf{X}$, to guarantee a solution $\boldsymbol{\theta}_S^\star$ satisfying, with high probability, the bound

$$\mathcal{L}(\boldsymbol{\theta}_S^\star) \leq C \cdot \mathcal{L}(\boldsymbol{\theta}^\star) + \mathcal{O}(\varepsilon), \qquad (4)$$

for some sufficiently large constant $C > 0$. They also argued that such additive error bounds are necessary, as achieving a pure multiplicative error bound would require exponentially many samples in $d$.

By using intricate tools from high-dimensional probability, subsequent work by Gajjar et al. (2024) improved the sample complexity to $\widetilde{\mathcal{O}}(d/\varepsilon^2)$, matching the linear benchmark up to polylogarithmic terms. Furthermore, they extended the analysis to the setting where $\phi$ is Lipschitz but unspecified by employing a meticulous "sampling-aware" discretization of the function class.

While the results in Gajjar et al. (2023; 2024) represent significant contributions toward achieving (2) for nonlinear models, they have a notable drawback: the guarantees in Gajjar et al. (2023; 2024) take the form (4), which involves an undesirable constant $C \gg 1$ (exceeding 1,000 in Gajjar et al. (2024)). Such large constants undermine the practical utility of the approximation guarantee in (4). We show that, under certain assumptions, it is possible to obtain the more desirable guarantee (2) (where $C = 1$).

## 3. Nonlinear Approximation: Our Approach

The core structure enabling the use of the subspace embedding property (3) in approximating linear least-squares problems is the linear *inner product*. Specifically, for $f(\boldsymbol{\theta}) = \langle \boldsymbol{\theta}, \mathbf{x} \rangle - y$, the least-squares model can be expressed as

$$\sum_{i=1}^{n} f_i^2(\boldsymbol{\theta}) = \sum_{i=1}^{n} (\langle \boldsymbol{\theta}, \mathbf{x}_i \rangle - y_i)^2 = \langle \widehat{\mathbf{X}} \widehat{\boldsymbol{\theta}}, \widehat{\mathbf{X}} \widehat{\boldsymbol{\theta}} \rangle,$$

where $\widehat{\mathbf{X}} = [\mathbf{X} \mid -\mathbf{y}] \in \mathbb{R}^{n \times (d+1)}$ and $\widehat{\boldsymbol{\theta}} = [\boldsymbol{\theta}; 1] \in \mathbb{R}^{d+1}$. This structure allows the subspace embedding property (3) to be applied, enabling the approximation of the original least-squares objective using a carefully chosen subsample of the data.

Using the notion of the *nonlinear adjoint operator*, we now derive an analogous "inner-product-like" representation for nonlinear mappings. This representation captures the non-linearity in a novel way, enabling the derivation of nonlinear embeddings similar to (3).

## 3.1. Nonlinear Adjoint Operator

Consider a mapping $f(.) : \mathbb{R}^p \to \mathbb{R}$ that is continuously differentiable. By the integral form of the mean value theorem,

$$f(\boldsymbol{\theta}) = f(\mathbf{0}) + \int_0^1 \left\langle \frac{\partial}{\partial \boldsymbol{\theta}} f(t\boldsymbol{\theta}), \boldsymbol{\theta} \right\rangle dt$$
$$= f(\mathbf{0}) + \left\langle \boldsymbol{\theta}, \int_0^1 \frac{\partial}{\partial \boldsymbol{\theta}} f(t\boldsymbol{\theta}) dt \right\rangle.$$

The above relation is an equation involving functions of $\boldsymbol{\theta}$. The second term includes an inner product between the parameter $\boldsymbol{\theta}$ and a vector-valued map, reminiscent of the role of dual operators in Banach spaces. This observation motivates the following definition, which extends beyond continuously differentiable functions.

**Definition 3.1** (Adjoint Operator). *The adjoint operator of $f : \mathbb{R}^p \to \mathbb{R}$ is the vector valued map $\mathbf{f}^\star : \mathbb{R}^p \to \mathbb{R}^p$,*

$$\mathbf{f}^\star(\boldsymbol{\theta}) \triangleq \int_0^1 \frac{\partial}{\partial \boldsymbol{\theta}} f(t\boldsymbol{\theta}) dt, \tag{5}$$

*for all $\boldsymbol{\theta} \in \mathbb{R}^p$ such that the function $\phi : [0,1] \to \mathbb{R}$ defined as $\phi(t) \triangleq f(t\boldsymbol{\theta})$ is absolutely continuous on $[0,1]$.*

Note that the absolute continuity condition relaxes the requirement for differentiability to that of being differentiable almost everywhere.

With the above definition, we can more compactly write

$$f(\boldsymbol{\theta}) = \left\langle \widehat{\boldsymbol{\theta}}, \widehat{\mathbf{f}}^\star(\boldsymbol{\theta}) \right\rangle, \tag{6a}$$

where

$$\widehat{\mathbf{f}}^\star(\boldsymbol{\theta}) \triangleq \begin{bmatrix} \mathbf{f}^\star(\boldsymbol{\theta}) \\ f(\mathbf{0}) \end{bmatrix}, \quad \text{and} \quad \widehat{\boldsymbol{\theta}} \triangleq \begin{bmatrix} \boldsymbol{\theta} \\ 1 \end{bmatrix}. \tag{6b}$$

The term nonlinear adjoint operator is motivated by cases where $f(\mathbf{0}) = 0$. In such cases, the above expression simplifies to $f(\boldsymbol{\theta}) = \langle \boldsymbol{\theta}, \mathbf{f}^\star(\boldsymbol{\theta}) \rangle$, which can be viewed as a nonlinear analogue of the Riesz representation theorem in linear settings; see, for example, Burýšková (1998); Scherpen & Gray (2002).

In the simplest case of linear regression, i.e., $f(\boldsymbol{\theta}) = \langle \boldsymbol{\theta}, \mathbf{x} \rangle - y$, we have $\mathbf{f}^\star(\boldsymbol{\theta}) = \mathbf{x}$, that is, the input data constitutes the space of adjoint operators. Beyond simple linear settings, the explicit calculation of (5) involves evaluating an integral. Naturally, this integral can be approximated using numerical methods, such as quadrature schemes. Fortunately, for many machine learning models, the following property enables the direct computation of (5).

**Proposition 3.1.** *Let $f = g \circ h = g(h)$ where $g : \mathbb{R} \to \mathbb{R}$ and $h : \mathbb{R}^p \to \mathbb{R}$. Also, assume that $h$ is positively homogeneous of degree $\alpha \in \mathbb{R}$, i.e., $h(t\boldsymbol{\theta}) = t^\alpha h(\boldsymbol{\theta})$ for any $t > 0$. Then*

$$\mathbf{f}^\star(\boldsymbol{\theta}) = \begin{cases} \left( \dfrac{g(h(\boldsymbol{\theta})) - g(0)}{\alpha(h(\boldsymbol{\theta}))} \right) \dfrac{\partial}{\partial \boldsymbol{\theta}} h(\boldsymbol{\theta}), & \text{if } h(\boldsymbol{\theta}) \neq 0, \\[2ex] \left( \dfrac{g'(0)}{\alpha} \right) \dfrac{\partial}{\partial \boldsymbol{\theta}} h(\boldsymbol{\theta}), & \text{if } h(\boldsymbol{\theta}) = 0. \end{cases}$$

*Proof.* The proof can be found in Appendix A.1. □

**Example 3.1** (Generalized Linear Predictors). *Consider generalized linear predictor models, often called "single index" models, where $f(\boldsymbol{\theta}) = \phi(\langle \boldsymbol{\theta}, \mathbf{x} \rangle)$ for some function $\phi : \mathbb{R} \to \mathbb{R}$ and data point $\mathbf{x} \in \mathbb{R}^d$. These models are foundational in studying complex nonlinear predictors in high dimensions (Hristache et al., 2001; Härdle et al., 2004; Kakade et al., 2011; Bietti et al., 2022). By limiting networks to a single hidden unit, they provide a controllable instance of neural network architecture while retaining simplicity (Goel et al., 2017). In scientific machine learning, such models underpin efficient surrogate methods for parametric partial differential equations, where the cost of labeled queries can be prohibitive (Cohen & DeVore, 2015; O'Leary-Roseberry et al., 2022).*

*Note that here $f(\boldsymbol{\theta}) = g(h(\boldsymbol{\theta}))$ where $g(t) = \phi(t)$ and $h(\boldsymbol{\theta}) = \langle \boldsymbol{\theta}, \mathbf{x} \rangle$. Since the positive homogeneity degree of $h$ is $\alpha = 1$, we have*

$$\mathbf{f}^\star(\boldsymbol{\theta}) = \left( \frac{\phi(\langle \boldsymbol{\theta}, \mathbf{x} \rangle) - \phi(0)}{\langle \boldsymbol{\theta}, \mathbf{x} \rangle} \right) \mathbf{x}. \tag{7}$$

*We also get $\widehat{\mathbf{f}}^\star(\boldsymbol{\theta}) = \begin{bmatrix} \mathbf{f}^\star(\boldsymbol{\theta}) \\ \phi(0) \end{bmatrix}$. A notable example is the logistic function where $\phi(t) = 1/(1 + \exp(-t))$, where*

$$\mathbf{f}^\star(\boldsymbol{\theta}) = - \left( \frac{\tanh(\langle \boldsymbol{\theta}, \mathbf{x} \rangle /2)}{2 \langle \boldsymbol{\theta}, \mathbf{x} \rangle} \right) \mathbf{x}.$$

**Example 3.2** (ReLU Neural Networks). *Suppose $r(\boldsymbol{\theta}) = \phi(\psi(\boldsymbol{\theta}))$ where $\phi : \mathbb{R} \to \mathbb{R}$, and $\psi(\boldsymbol{\theta}) \triangleq a \cdot \max\{\langle \mathbf{b}, \mathbf{x} \rangle, 0\})$ for $\mathbf{x} \in \mathbb{R}^d$ and $\boldsymbol{\theta} = [a, \mathbf{b}] \in \mathbb{R}^{d+1}$. Here, the positive homogeneity degree of $\psi$ is $\alpha = 2$. Hence,*

$$r^\star(\boldsymbol{\theta}) = \int_0^1 \frac{\partial}{\partial \boldsymbol{\theta}} \phi(ta \cdot \max\{\langle t\mathbf{b}, \mathbf{x} \rangle, 0\}) \, dt \tag{8}$$

$$= \frac{\phi(a \cdot \max\{\langle \mathbf{b}, \mathbf{x} \rangle, 0\}) - \phi(0)}{2a \cdot \max\{\langle \mathbf{b}, \mathbf{x} \rangle, 0\}} \begin{bmatrix} \max\{\langle \mathbf{b}, \mathbf{x} \rangle, 0\} \\ a \cdot \mathbf{x} \cdot \mathbb{1}_{\{\langle \mathbf{b}, \mathbf{x} \rangle > 0\}} \end{bmatrix}.$$

*In the typical case where $\phi(z) = z$, this simplifies to*

$$r^\star(\boldsymbol{\theta}) = \frac{1}{2} \begin{bmatrix} \max\{\langle \mathbf{b}, \mathbf{x} \rangle, 0\} \\ a \cdot \mathbf{x} \mathbb{1}_{\{\langle \mathbf{b}, \mathbf{x} \rangle > 0\}} \end{bmatrix}.$$

*Now, consider a two-layer neural network with $m$ hidden neurons, ReLU activation, and a single output:*

$$f(\boldsymbol{\theta}) = \sum_{j=1}^{m} r(\boldsymbol{\theta}_j) = \sum_{j=1}^{m} \phi(a_j \cdot \max\{\langle \mathbf{b}_j, \mathbf{x} \rangle, 0\}),$$

*where $\boldsymbol{\theta}_j = [a_j, \mathbf{b}_j]$ and $\boldsymbol{\theta} = [\boldsymbol{\theta}_1, \ldots, \boldsymbol{\theta}_m]$. We have*

$$\mathbf{f}^\star(\boldsymbol{\theta}) = \sum_{j=1}^{m} \int_0^1 \frac{\partial}{\partial \boldsymbol{\theta}} \phi(t a_j \cdot \max\{\langle t\mathbf{b}_j, \mathbf{x} \rangle, 0\}) \mathrm{d}t$$

$$= \sum_{j=1}^{m} \mathbf{e}_j \otimes \mathbf{r}^\star(\boldsymbol{\theta}_j)$$

$$= \left[ [\mathbf{r}^\star(\boldsymbol{\theta}_1)]^{\mathsf{T}} \quad [\mathbf{r}^\star(\boldsymbol{\theta}_2)]^{\mathsf{T}} \quad \ldots \quad [\mathbf{r}^\star(\boldsymbol{\theta}_m)]^{\mathsf{T}}, \right]^{\mathsf{T}}$$

*where $\otimes$ denotes the Kronecker product and $\mathbf{r}^\star(\boldsymbol{\theta}_j)$ is given in (8). Finally, we get $\widehat{\mathbf{f}}^\star(\boldsymbol{\theta}) = \begin{bmatrix} \mathbf{f}^\star(\boldsymbol{\theta}) \\ m \cdot \phi(0) \end{bmatrix}$.*

### 3.2. Parameter-dependent Approximations

The "inner-product-like" representation of a nonlinear function in (6) enables the development of sampling strategies for approximating general loss functions beyond linear least-squares and extends the concept of linear subspace embeddings to nonlinear settings. We focus on nonlinear least-squares problems, where $\ell(t) = t^2$, and (1) is expressed as $\mathcal{L}(\boldsymbol{\theta}) = \sum_{i=1}^{n} (f_i(\boldsymbol{\theta}))^2$, with extensions to more general nonlinear losses discussed in Appendix A.3.

**Definition 3.2** (Nonlinear Dual Matrix). *Given the nonlinear maps $f_i : \mathbb{R}^p \to \mathbb{R}$, $i = 1, \ldots, n$, the nonlinear dual matrix operator, $\mathbf{F}^\star : \mathbb{R}^p \to \mathbb{R}^{n \times p}$, is defined as*

$$\mathbf{F}^\star(\boldsymbol{\theta}) \triangleq \begin{bmatrix} \mathbf{f}_1^\star(\boldsymbol{\theta}) & \mathbf{f}_2^\star(\boldsymbol{\theta}) & \ldots & \mathbf{f}_n^\star(\boldsymbol{\theta}) \end{bmatrix}^{\mathsf{T}}.$$

*Using (6b), we also define*

$$\widehat{\mathbf{F}}^\star(\boldsymbol{\theta}) \triangleq \begin{bmatrix} \widehat{\mathbf{f}}_1^\star(\boldsymbol{\theta}) & \widehat{\mathbf{f}}_2^\star(\boldsymbol{\theta}) & \ldots & \widehat{\mathbf{f}}_n^\star(\boldsymbol{\theta}) \end{bmatrix}^{\mathsf{T}}.$$

With (6) and Definition 3.2, we can now write

$$\mathcal{L}(\boldsymbol{\theta}) = \sum_{i=1}^{n} (f_i(\boldsymbol{\theta}))^2 = \sum_{i=1}^{n} \langle \widehat{\boldsymbol{\theta}}, \widehat{\mathbf{f}}_i^\star(\boldsymbol{\theta}) \rangle^2 = \left\| \widehat{\mathbf{F}}^\star(\boldsymbol{\theta}) \widehat{\boldsymbol{\theta}} \right\|^2. \quad (9)$$

This formulation resembles the ordinary least-squares setup, with the nonlinear dual matrix $\widehat{\mathbf{F}}^\star(\boldsymbol{\theta})$ replacing the original input data matrix. This motivates the following notions of importance sampling scores, which generalize standard leverage scores or row-norms to those based on the rows of the dual matrix $\widehat{\mathbf{F}}^\star(\boldsymbol{\theta})$.

**Definition 3.3** (Nonlinear Leverage Scores). *Nonlinear leverage score of $f_i$ is defined as*

$$\tau_i(\boldsymbol{\theta}) \triangleq \frac{\left\langle \mathbf{e}_i, \widehat{\mathbf{F}}^\star(\boldsymbol{\theta}) \left[ \widehat{\mathbf{F}}^\star(\boldsymbol{\theta}) \right]^\dagger \mathbf{e}_i \right\rangle}{\mathrm{Rank}\left( \widehat{\mathbf{F}}^\star(\boldsymbol{\theta}) \right)}.$$

**Definition 3.4** (Nonlinear Norm Scores). *Nonlinear norm score of $f_i$ is defined as*

$$\tau_i(\boldsymbol{\theta}) \triangleq \frac{\left\| \widehat{\mathbf{f}}_i^\star(\boldsymbol{\theta}) \right\|_2^2}{\left\| \widehat{\mathbf{F}}^\star(\boldsymbol{\theta}) \right\|_{\mathrm{F}}^2}.$$

It is easy to verify that, for both Definitions 3.3 and 3.4, the values $\{\tau_i(\boldsymbol{\theta})\}_{i=1}^n$ form a probability distribution, i.e., $\tau_i(\boldsymbol{\theta}) \geq 0$ and $\sum_i \tau_i(\boldsymbol{\theta}) = 1$.

**Remark 3.1.** *Definitions 3.3 and 3.4 generalize their linear counterparts. Specifically, for linear models where $f(\boldsymbol{\theta}) = \langle \boldsymbol{\theta}, \mathbf{x} \rangle$, these definitions reduce to the standard linear leverage and row norm scores. To the best of our knowledge, this represents the first systematic extension of these concepts from linear to nonlinear settings.*

**Remark 3.2.** *While Example 3.1 and Example 3.2 illustrate explicit computations of the adjoint operator via Proposition 3.1, our approach conceptually extends to broader classes of nonlinear functions through its general Definition 3.1. When an explicit adjoint form is available, importance scores (row-norm or leverage) are computed directly from the nonlinear dual matrix via standard linear algebra routines (e.g., QR or SVD factorizations). In scenarios where the adjoint lacks a closed-form expression, we can approximate it numerically using Definition 3.1; the resulting dual matrix is then factorized similarly, with QR or SVD methods.*

Let $\mathcal{S}$ be the index set of $s$ samples, obtained by randomly sampling $s \leq n$ data points with replacement, using probabilities defined by one of the sampling distributions in Definitions 3.3 and 3.4. Note that $\mathcal{S}$ depends on the choice of $\boldsymbol{\theta}$ (this is addressed later in Section 3.3.1). The loss on the sampled data is defined as

$$\mathcal{L}_{\mathcal{S}}(\boldsymbol{\theta}) \triangleq \sum_{i \in \mathcal{S}} \frac{(f_i(\boldsymbol{\theta}))^2}{s \tau_i(\boldsymbol{\theta})},$$

which is an unbiased estimator of the true loss. Similar to the derivation of (9), we have

$$\mathcal{L}_{\mathcal{S}}(\boldsymbol{\theta}) = \|\widehat{\mathbf{F}}_{\mathcal{S}}^\star(\boldsymbol{\theta}) \widehat{\boldsymbol{\theta}}\|^2,$$

where $\widehat{\mathbf{F}}_{\mathcal{S}}^\star(\boldsymbol{\theta}) \in \mathbb{R}^{s \times p}$ is a subset of $\widehat{\mathbf{F}}^\star(\boldsymbol{\theta})$, consisting of rows indexed by $\mathcal{S}$, with the $i$th row scaled by $1/\sqrt{s \tau_i(\boldsymbol{\theta})}$.

The representation in (9), which related the output of a nonlinear function to the norm of a matrix-vector product, enables the use of existing results on approximate matrix multiplication and linear subspace embedding to derive an approximation bound on $\mathcal{L}_{\mathcal{S}}(\boldsymbol{\theta})$; see, for example, Woodruff et al. (2014); Drineas & Mahoney (2018); Martinsson & Tropp (2020). Specifically, for a fixed $\boldsymbol{\theta}$ and

any of the sampling distributions mentioned, if the sample size is $\mathcal{O}\left(p\log(p/\delta)/\varepsilon^2\right)$, then with probability at least $1-\delta$, for all $\mathbf{v} \in \mathbb{R}^{p+1}$, $(1-\varepsilon)\|\widehat{\mathbf{F}}^\star(\boldsymbol{\theta})\mathbf{v}\|^2 \leq \|\widehat{\mathbf{F}}_{\mathcal{S}}^\star(\boldsymbol{\theta})\mathbf{v}\|^2 \leq (1+\varepsilon)\|\widehat{\mathbf{F}}^\star(\boldsymbol{\theta})\mathbf{v}\|^2$. which in turn implies, with probability at least $1-\delta$,

$$(1-\varepsilon)\mathcal{L}(\boldsymbol{\theta}) \leq \mathcal{L}_{\mathcal{S}}(\boldsymbol{\theta}) \leq (1+\varepsilon)\mathcal{L}(\boldsymbol{\theta}). \tag{10}$$

### 3.3. Parameter-independent Approximation

The relation (10) and the optimality of $\boldsymbol{\theta}_{\mathcal{S}}^\star$ with respect to $\mathcal{L}_{\mathcal{S}}(.)$ allow us to take initial steps towards (2) as

$$\mathcal{L}_{\mathcal{S}}(\boldsymbol{\theta}_{\mathcal{S}}^\star) \leq \mathcal{L}_{\mathcal{S}}(\boldsymbol{\theta}^\star) \leq (1+\varepsilon)\mathcal{L}(\boldsymbol{\theta}^\star). \tag{11}$$

However, the key missing component is that the left-hand side of (2) involves the loss evaluated over the full dataset, whereas the left-hand side of (11) is restricted to the loss over the sampled subset. The chain of inequalities to obtain (2) would be complete if we could find a sensible lower bound for $\mathcal{L}_{\mathcal{S}}(\boldsymbol{\theta}_{\mathcal{S}}^\star)$ in terms of $\mathcal{L}(\boldsymbol{\theta}_{\mathcal{S}}^\star)$. Additionally, (11) requires evaluating the importance sampling scores defined in Definitions 3.3 and 3.4 at $\boldsymbol{\theta}^\star$, which is inherently unknown. Therefore, to achieve (2), we must address two critical challenges:

1. Approximating the nonlinear scores defined in Definitions 3.3 and 3.4 with quantities that are independent of the parameter $\boldsymbol{\theta}$.

2. Establishing a meaningful lower bound for $\mathcal{L}_{\mathcal{S}}(\boldsymbol{\theta}_{\mathcal{S}}^\star)$ in terms of $\mathcal{L}(\boldsymbol{\theta}_{\mathcal{S}}^\star)$.

Our next tasks involve addressing these challenges.

#### 3.3.1. ESTIMATING SAMPLING SCORES

The relation in (11) requires calculating the scores from Definitions 3.3 and 3.4 for $\boldsymbol{\theta}^\star$, which is infeasible since $\boldsymbol{\theta}^\star$ is unknown. A solution is to approximate these nonlinear scores with ones independent of $\boldsymbol{\theta}$.

Sampling with near-optimal probabilities has been studied in RandNLA (Drineas & Mahoney, 2018; Woodruff et al., 2014), where it is known that if leverage or row-norm scores, $\{\tau_i\}$, are approximated by $\{\hat{\tau}_i\}$ such that $\beta\tau_i \leq \hat{\tau}_i$, the subspace embedding property (3) holds with sample size $\mathcal{O}(p\log(p/\delta)/(\beta\varepsilon^2))$. If nonlinear scores can be similarly approximated in a manner independent of $\boldsymbol{\theta}$, we can sample according to the approximate scores and still obtain a guarantee of the form (10) for any $\boldsymbol{\theta}^\star$.

Within the context of the specific models considered in Examples 3.1 and 3.2, we demonstrate that leveraging the structure of the nonlinear adjoint operators enables the estimation of the nonlinear scores.

**Example 3.3** (Generalized Linear Predictors). *Consider the class of generalized linear predictor models from Example 3.1 with any activation function $\phi$ such that there exist*

constants $0 < l < u < \infty$ *for which* $l \leq \phi^2(t)/t^2 \leq u$ *for all* $t \in \mathcal{T}$, *where* $\mathcal{T}$ *is some set of interest. An example is the Swish-type activation function given by*

$$\phi(t) = t \cdot \left(\sqrt{c_1} + (\sqrt{c_2} - \sqrt{c_1})\frac{1}{1 + e^{-\zeta t}}\right),$$

*for some* $0 \leq c_1 < c_2$ *and* $\zeta \in \mathbb{R}$. *If* $c_1 > 0$, *we can take* $l = \sqrt{c_1}$ *and* $u = \sqrt{c_2}$. *For* $c_1 = 0$, *i.e., the typical Swish function, we can still define* $0 < l \triangleq \min_{t \in \mathcal{T}} \phi^2(t)/t^2$ *provided that* $\mathcal{T}$ *is a bounded set.*

*Consider leverage score sampling according to Definition 3.3. Suppose* $\mathbf{X} \in \mathbb{R}^{n \times d}$ *and* $\mathbf{F}^\star(\boldsymbol{\theta}) \in \mathbb{R}^{n \times d}$ *are both full column rank. Since*

$$[\mathbf{F}^\star(\boldsymbol{\theta})]^\mathsf{T}\mathbf{F}^\star(\boldsymbol{\theta}) = \sum_i \left(\frac{\phi(\langle\boldsymbol{\theta}, \mathbf{x}_i\rangle)}{\langle\boldsymbol{\theta}, x\rangle}\right)^2 \mathbf{x}_i\mathbf{x}_i^T,$$

*it follows that, as long as* $c_1 > 0$ *or* $\boldsymbol{\theta} \in \mathcal{C}$ *for some bounded set* $\mathcal{C}$, *we have* $\mathbf{0} \prec l \cdot \mathbf{X}^\mathsf{T}\mathbf{X} \preceq [\mathbf{F}^\star(\boldsymbol{\theta})]^\mathsf{T}\mathbf{F}^\star(\boldsymbol{\theta}) \preceq u \cdot \mathbf{X}^\mathsf{T}\mathbf{X}$. *Hence,*

$$\begin{aligned}
d \cdot \tau_i(\boldsymbol{\theta}) &= \left\langle \mathbf{f}^\star(\boldsymbol{\theta}), \left([\mathbf{F}^\star(\boldsymbol{\theta})]^\mathsf{T}\mathbf{F}^\star(\boldsymbol{\theta})\right)^{-1}\mathbf{f}^\star(\boldsymbol{\theta})\right\rangle \\
&\leq \frac{1}{c_1}\left\langle \mathbf{f}^\star(\boldsymbol{\theta}), (\mathbf{X}^\mathsf{T}\mathbf{X})^{-1}\mathbf{f}^\star(\boldsymbol{\theta})\right\rangle \\
&\leq \frac{u}{l}\left\langle \mathbf{x}, (\mathbf{X}^\mathsf{T}\mathbf{X})^{-1}\mathbf{x}\right\rangle = \frac{d \cdot u}{l}\tau_i,
\end{aligned}$$

*where* $\tau_i$ *is the linear leverage score. This implies* $\beta = l/u$.

**Example 3.4** (ReLU Neural Networks). *Revising the NN model in Example 3.2, we consider row norm sampling according to Definition 3.4. Suppose* $\phi$ *is such that such that* $c_1 \leq (\phi(t) - \phi(0))^2/t^2 \leq c_2$ *for some* $0 < c_1 \leq c_2 < \infty$ *and for all* $t \in \mathcal{T}$ *for some set of interest* $\mathcal{T}$, *e.g., Swish-type or linear output layer. Recall that* $\boldsymbol{\theta} = [\boldsymbol{\theta}_1, \ldots, \boldsymbol{\theta}_m]$ *where* $\boldsymbol{\theta}_j = [a_j, \mathbf{b}_j]$. *Also denote* $\boldsymbol{\theta}^\star = [\boldsymbol{\theta}_1^\star, \ldots, \boldsymbol{\theta}_m^\star]$ *where* $\boldsymbol{\theta}_j^\star = [a_j^\star, \mathbf{b}_j^\star]$. *Suppose* $a_j^\star \neq 0$ *for all* $j \in 1, 2, \ldots, m$. *Let* $l > 0$ *be such that* $\min_j(a_j^\star)^2 \geq l$ *and* $0 < u < \infty$ *be such that* $\sum_{j=1}^m \left(\|\mathbf{b}_j^\star\|^2 + (a_j^\star)^2\right) \leq u$. *Define the set*

$$\mathcal{C} \triangleq \Bigg\{[a_1, \mathbf{b}_1, \ldots, a_m, \mathbf{b}_m] \mid \min_{j=1,\ldots,m}(a_j)^2 \geq l,$$

$$\sum_{j=1}^m \left(\|\mathbf{b}_j\|^2 + (a_j)^2\right) \leq u\Bigg\}.$$

*Clearly, by construction,* $\boldsymbol{\theta}^\star \in \mathcal{C}$. *After some algebraic manipulations (see Appendix A.2), for any* $\mathbf{x}_i$ *and any* $\boldsymbol{\theta} \in \mathcal{C}$, *we have*

$$c_1 l \|\mathbf{x}_i\|^2 \leq \|\mathbf{f}_i^\star(\boldsymbol{\theta})\|^2 \leq c_2 u \|\mathbf{x}_i\|^2,$$

*where* $\mathbf{f}_i^\star(\boldsymbol{\theta})$ *is the adjoint operator of*

$$f_i(\boldsymbol{\theta}) = \sum_{j=1}^m \phi_i(a_j \cdot \max\{\langle \mathbf{b}_j, \mathbf{x}_i \rangle, 0\}).$$

*This implies that*

$$\tau_i(\boldsymbol{\theta}) = \frac{\left\|\widehat{\mathbf{f}}_i^\star(\boldsymbol{\theta})\right\|_2^2}{\left\|\widehat{\mathbf{F}}^\star(\boldsymbol{\theta})\right\|_F^2} \le \left(\frac{\max\{c_2 u, 1\}}{\min\{c_1 l, 1\}}\right) \tau_i,$$

*where* $\tau_i$ *is the norm score for the* $i^{th}$ *row of*

$$\widehat{\mathbf{X}} = \begin{bmatrix} \mathbf{x}_1 & \mathbf{x}_2 & \dots & \mathbf{x}_n \\ m\phi_1(0) & m\phi_2(0) & \dots & m\phi_n(0) \end{bmatrix}^{\mathsf{T}} \in \mathbb{R}^{n \times (d+1)}.$$

*This implies* $\beta = \min\{c_1 l, 1\}/\max\{c_2 u, 1\}$.

### 3.3.2. LOWER BOUNDING $\mathcal{L}_{\mathcal{S}}(\boldsymbol{\theta}_{\mathcal{S}}^\star)$

Let $\boldsymbol{\theta}^\star$ denote a solution to (1), and consider a ball with radius $R$, chosen large enough to contain $\boldsymbol{\theta}^\star$, denoted by $\mathcal{B}_R^\star$. For any $\varepsilon \in (0, 1)$, we pick a discrete subset $\mathcal{N}_\varepsilon \subseteq \mathcal{B}_R^\star$ such that, for every $\boldsymbol{\theta} \in \mathcal{B}_R^\star$, there exists at least one $\boldsymbol{\theta}' \in \mathcal{N}_\varepsilon$ satisfying $\|\boldsymbol{\theta} - \boldsymbol{\theta}'\|_2 \le \varepsilon R$. The size of this set can be shown to be $|\mathcal{N}_\varepsilon| \in \Omega(1/\varepsilon^p)$. This construction is well-known and is commonly referred to as an $\varepsilon$-net; see Appendix A.4.

This $\varepsilon$-net construction enables a union bound argument to control the probability of (10) for all $\boldsymbol{\theta} \in \mathcal{N}_\varepsilon$. Suppose $\mathcal{S}$ can be sampled according to a uniform estimate of the nonlinear scores, as described in Section 3.3.1, so that $\mathcal{S}$ no longer depends on the choice of $\boldsymbol{\theta}$. If, for a given $\boldsymbol{\theta}$, (10) fails to hold with a probability of at most $\delta'$, then a union bound ensures that the overall failure probability of (10) for all $\boldsymbol{\theta} \in \mathcal{N}_\varepsilon$ cannot exceed $|\mathcal{N}_\varepsilon|\delta'$. By choosing $\delta' = \delta/|\mathcal{N}_\varepsilon| \le \delta\varepsilon^p$, we ensure (10) holds with a success probability of at least $1 - \delta$ for all $\boldsymbol{\theta} \in \mathcal{N}_\varepsilon$.

Suppose $\mathcal{L}_{\mathcal{S}}(.)$ is continuous on $\mathcal{B}_R^\star$, which is almost always guaranteed for many loss functions in ML. The compactness of the ball implies that $\mathcal{L}_{\mathcal{S}}(.)$ is also Lipchitz continuous with respect to $\boldsymbol{\theta}$ on $\mathcal{B}_R^\star$, i.e., for any set of samples $\mathcal{S}$, there exists a constant $0 \le L(f, \mathbf{X}, \mathcal{S}, R) < \infty$ such that for any $\boldsymbol{\theta}, \boldsymbol{\theta}' \in \mathcal{B}_R^\star, |\mathcal{L}_{\mathcal{S}}(\boldsymbol{\theta}) - \mathcal{L}_{\mathcal{S}}(\boldsymbol{\theta}')| \le L(f, \mathbf{X}, \mathcal{S}, R)\|\boldsymbol{\theta} - \boldsymbol{\theta}'\|_2$. Note that the Lipschitz continuity constant may depend on $\mathbf{X}, \mathcal{S}, f$, and the radius of the ball $\mathcal{B}_R^\star$. Define $L(f, \mathbf{X}, R) \triangleq \max_{\mathcal{S}} L(f, \mathbf{X}, \mathcal{S}, R)$. Note that this is a maximization over a finite collection of numbers[1] and hence $L(f, \mathbf{X}, R) < \infty$.

Now, suppose $\mathcal{C} \subseteq \mathcal{B}_R^\star$ with $\boldsymbol{\theta}^\star \in \mathcal{C}$, and let

$$\boldsymbol{\theta}_{\mathcal{S}}^\star \in \arg\min_{\boldsymbol{\theta} \in \mathcal{C}} \mathcal{L}_{\mathcal{S}}(\boldsymbol{\theta}). \tag{12}$$

---

[1]The total number of possible unordered samples of any size from $n$ object with replacement is given by $\sum_{s=1}^n \binom{n+s-1}{s}$.

Also, let $\boldsymbol{\theta}_0 \in \mathcal{N}_\varepsilon$ be such that $\|\boldsymbol{\theta}_{\mathcal{S}}^\star - \boldsymbol{\theta}_0\|_2 \le \varepsilon R$. Without loss of generality, we can assume $\boldsymbol{\theta}_0 \in \mathcal{N}_\varepsilon \cap \mathcal{C}$ as otherwise we can simply increase the size of the net inside $\mathcal{C}$ by $\mathcal{O}(|\mathcal{N}_\varepsilon|)$. From Lipschitz continuity and (10), we get

$$\mathcal{L}(\boldsymbol{\theta}_{\mathcal{S}}^\star) \le \mathcal{L}(\boldsymbol{\theta}_0) + \varepsilon R \cdot L(f, \mathbf{X}, R)$$
$$\le \frac{1}{1-\varepsilon}\mathcal{L}_{\mathcal{S}}(\boldsymbol{\theta}_0) + \varepsilon R \cdot L(f, \mathbf{X}, R)$$
$$\le \frac{1}{1-\varepsilon}\Big(\mathcal{L}_{\mathcal{S}}(\boldsymbol{\theta}_{\mathcal{S}}^\star) + \varepsilon R \cdot L(f, \mathbf{X}, R)\Big) + \varepsilon R \cdot L(f, \mathbf{X}, R),$$

which gives the lower bound

$$\mathcal{L}_{\mathcal{S}}(\boldsymbol{\theta}_{\mathcal{S}}^\star) \ge (1-\varepsilon)\mathcal{L}(\boldsymbol{\theta}_{\mathcal{S}}^\star) - \varepsilon(2-\varepsilon)R \cdot L(f, \mathbf{X}, R).$$

Combining this with (11), we get

$$\mathcal{L}(\boldsymbol{\theta}_{\mathcal{S}}^\star) \le \mathcal{L}(\boldsymbol{\theta}^\star) + \frac{\varepsilon}{1-\varepsilon}\Big(\mathcal{L}(\boldsymbol{\theta}^\star) + (2-\varepsilon)R \cdot L(f, \mathbf{X}, R)\Big).$$

The culmination of the above discussions and derivations leads to the main result of this paper.

**Theorem 3.1** (Main Result)**.** *Suppose* $\boldsymbol{\theta}^\star \in \mathcal{C}$ *is a solution to* (1) *with squared loss* $\ell(t) = t^2$ *and for some* $0 < \beta \le 1$, *we have* $\beta\tau_i(\boldsymbol{\theta}) \le \tau_i$ *for* $i = 1, 2, \dots, n$ *and for all* $\boldsymbol{\theta} \in \mathcal{C}$. *Consider random sampling according to* $\tau_i$ *with a sample* $\mathcal{S}$ *of size at least* $\mathcal{O}\left(\left(p\log(p/\delta) + p^2\log(p/\varepsilon)\right)/(\beta\varepsilon^2)\right)$ *for some* $\varepsilon \in (0, 1)$ *and* $\delta \in (0, 1)$. *Let* $\boldsymbol{\theta}_{\mathcal{S}}^\star$ *be defined as in* (12). *Then* (2) *holds with probability at least* $1 - \delta$.

**Remark 3.3.** *Theorem 3.1 applies to more generally than single index models. However, when adapted to Example 3.3, it provides a result similar to Gajjar et al. (2024). While both works have the same dependence on* $\varepsilon$, *our result has quadratic dependence on the dimension, while theirs is linear. Additionally, our approach involves a constrained optimization subproblem* (12), *whereas the subproblems of Gajjar et al. (2024) are unconstrained. In contrast, Gajjar et al. (2024) provided a guarantee of the form* (4) *with a potentially large constant* $C \gg 1$, *while Theorem 3.1 provides a more desirable guarantee of the form* (2).

## 4. Experiments

We conduct a series of experiments to demonstrate the effectiveness and versatility of the proposed nonlinear importance scores. First, we evaluate our approach on several benchmarking regression datasets, including California Housing Prices (Pace & Barry, 1997), Medical Insurance dataset (Lantz, 2019), and Diamonds dataset (Wickham & Sievert, 2009). As a proof of concept, we demonstrate that our importance sampling method outperforms traditional linear and uniform sampling strategies by achieving lower training error using fewer samples. Second, we consider image classification using four standard datasets: SVHN (Street View House Numbers) (Netzer et al., 2011), FER-2013

(Facial Expression Recognition) (Goodfellow et al., 2013), NOTMNIST (Bulatov, 2011), and QD (Quick, Draw) (Ha & Eck, 2018). Here, we demonstrate that our proposed nonlinear leverage scores in Definition 3.3 serve as a powerful diagnostic tool, identifying the most "important" points in training complex models and detecting anomalies by isolating outliers. We employ both single-index models and neural networks to validate our findings across varying model complexities and architectures. Additional experimental details are provided in Appendix A.5. The code is available here.

**Regression Tasks.** We compare the efficacy of three sampling methods: uniform sampling, linear importance sampling (based on leverage scores and row-norms of the original dataset), and nonlinear importance sampling as defined in Definitions 3.3 and 3.4. While computing nonlinear importance scores directly is impractical in real-world scenarios, these experiments serve to validate that such scores effectively identify critical samples. As shown in Figure 1, nonlinear importance sampling more effectively identifies impactful samples, achieving lower MSE with fewer training points compared to uniform and linear sampling. The Y-axis for regression tasks reports relative error on a logarithmic scale, so even small vertical shifts represent substantial absolute improvements.

**Classification Tasks.** We convert the datasets into binary classification tasks, comparing visually similar ("like") and dissimilar ("unlike") classes, in SVHN and NotMNIST. "like" classes are those that are harder to distinguish without prior knowledge, while "unlike" classes are more easily separable. In SVHN, we examine two scenarios: distinguishing the clearly distinct digits "1" vs. "0" and the more visually similar "1" vs. "7." Similarly, in NotMNIST, we consider letter pairs (A vs. B) and (B vs. D). We also analyze more complex, noisy datasets like QD, where we distinguish between "Fish" and "Car." For FER, which groups expressions by valence (positive vs. negative) based on (Mollahosseini et al., 2017), we track the evolution of important samples as the model approaches optimal parameters ($\approx 55\%$ at initialization, $\approx 65\%$, and $\approx 75\%$ accuracy at termination).

Figure 2 presents images with the highest (most unique) and lowest (easiest to classify) nonlinear leverage scores for each binary task. The results clearly show that samples with higher nonlinear leverage scores contain distinct patterns and are harder to classify, while those with lower scores are straightforward. In contrast, standard linear leverage scores select more random and less meaningful/insightful samples. Notably, our method identifies mislabeled and noisy samples with high scores, which represent the outliers. In FER, under-trained models highlight blank or extreme valence, whereas trained models detect subtler emotions and facial characteristics such as accessories, tears, and aging. To demonstrate that nonlinear scores also yield robust

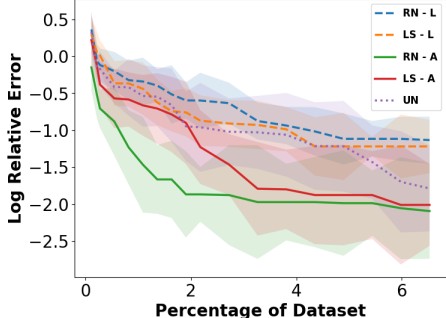

(a) California Housing Dataset

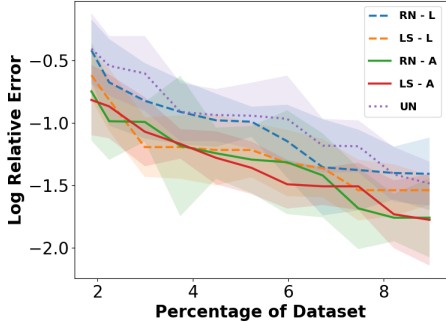

(b) Medical Insurance Dataset

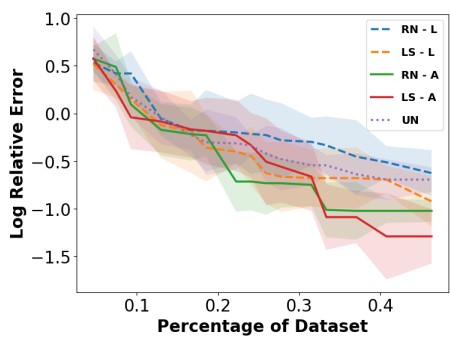

(c) Diamonds dataset

*Figure 1.* Comparison of sampling strategies. The Y-axis shows $\log\left[(\mathcal{L}(\boldsymbol{\theta}_{\mathcal{S}}^{\star}) - \mathcal{L}(\boldsymbol{\theta}^{\star}))/\mathcal{L}(\boldsymbol{\theta}^{\star})\right]$ against sample size (as a percentage of total data), where $\boldsymbol{\theta}_{\mathcal{S}}^{\star}$ is the optimal parameter from training on sampled data. "RN", "LS", and "UN" denote Row Norm, Leverage Scores, and Uniform Sampling, respectively, with "L" and "A" indicating linear and adjoint-based nonlinear variants. Nonlinear importance scores consistently outperform all other alternatives.

quantitative evidence for our classification experiments, we provide additional numerical results in Appendix A.6. Additional images supporting Figure 2 for all datasets are given in Appendix A.7.

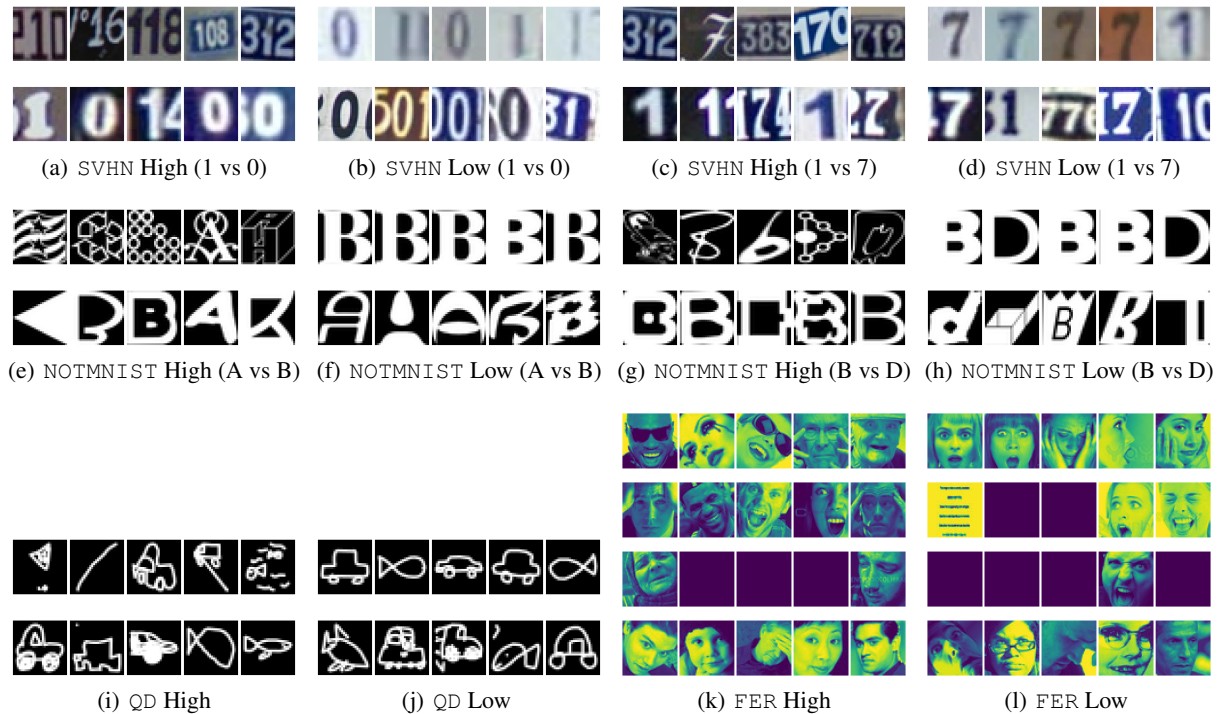

(a) SVHN High (1 vs 0)    (b) SVHN Low (1 vs 0)    (c) SVHN High (1 vs 7)    (d) SVHN Low (1 vs 7)

(e) NOTMNIST High (A vs B)    (f) NOTMNIST Low (A vs B)    (g) NOTMNIST High (B vs D)    (h) NOTMNIST Low (B vs D)

(i) QD High    (j) QD Low    (k) FER High    (l) FER Low

*Figure 2.* Comparisons of high and low linear/nonlinear leverage scores across multiple datasets. "High" and "Low" refer to images with the highest and lowest scores, respectively. In subfigures (a)-(j), the top row shows images selected using nonlinear leverage scores (Definition 3.3), while the bottom row uses linear leverage scores. Samples with higher nonlinear scores contain distinct patterns and are harder to classify, while lower scores correspond to straightforward samples. In contrast, linear scores select less insightful samples. Notably, our method identifies mislabeled and noisy samples (outliers) with high scores. In FER, the top three rows represent the scores calculated at $\approx 75\%$ (final), $\approx 65\%$, and $\approx 55\%$ (initial) training accuracy, while the bottom row uses linear scores. Under-trained models highlight extreme expressions, while fully trained models detect subtler emotions and facial features like accessories, tears, and aging.

## 5. Conclusions and Further Thoughts

We introduced a unifying framework that extends importance sampling from linear to more general nonlinear models, through the notion of the adjoint operator of a nonlinear map. This perspective yields sampling schemes with approximation guarantees analogous to linear subspace embeddings, yet it is applicable to a wide range of models. Our theoretical analysis shows that these generalized scores offer strong performance bounds and our experiments demonstrate concrete benefits in reduced training costs, improved diagnostics, and outlier detection.

While we did not explicitly address active learning or transfer learning scenarios in this paper, our nonlinear importance sampling approach can naturally be utilized in such contexts. In particular, the method of subsampling based on nonlinear scores can be viewed as a one-shot active learning strategy, where selecting the most informative samples significantly reduces labeling costs. Similarly, these nonlinear scores could also be beneficial in analyzing transfer learning, by effectively identifying samples from a source domain that best represent the characteristics of a target domain.

Although Appendix A.3 outlines preliminary steps toward extending the theoretical guarantees beyond squared loss objectives, fully generalizing them remains an avenue for future work. By estimating nonlinear importance scores without the model parameters, we substantially reduce the classic "chicken-and-egg" coupling between sampling and estimation. The subsequent constrained optimization still introduces a mild dependence on the unknown solution, leaving open challenges for fully parameter-agnostic importance sampling in nonlinear settings.

As shown, in many cases the computational cost of approximating our nonlinear importance scores is comparable to that of existing sampling techniques in the linear regime, as both rely on similar fundamental operations. In such cases, the only additional overhead typically stems from computing the nonlinear dual matrix. Thus, our framework in many cases does not introduce any fundamentally new computational bottleneck beyond those already present in linear importance sampling methods. As a result, the practical runtime overhead is largely governed by implementation-level factors, such as the choice of optimization algorithm and hyperparameter settings.

## Acknowledgments

We sincerely thank Cameron Musco for their valuable feedback and discussions. This research was partially supported by the Australian Research Council through an Industrial Transformation Training Centre for Information Resilience (IC200100022).

## Impact Statement

This paper presents work whose goal is to advance the field of Machine Learning. There are many potential societal consequences of our work, none which we feel must be specifically highlighted here.

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

# A. Appendix

### A.1. Proof of Proposition 3.1

*Proof.* We first note that by Euler's homogeneous function theorem, the gradient of $h$ is positively homogeneous of degree $\alpha - 1$, i.e.,

$$\frac{\partial}{\partial \boldsymbol{\theta}} h(t\boldsymbol{\theta}) = t^{\alpha - 1} \frac{\partial}{\partial \boldsymbol{\theta}} h(\boldsymbol{\theta}), \quad \forall t > 0.$$

We have,

$$
\begin{aligned}
\mathbf{f}^{\star}(\boldsymbol{\theta}) &= \int_0^1 \frac{\partial}{\partial \boldsymbol{\theta}} f(t\boldsymbol{\theta}) \mathrm{d}t = \int_0^1 \frac{\partial}{\partial \boldsymbol{\theta}} h(t\boldsymbol{\theta}) g'(h(t\boldsymbol{\theta})) \mathrm{d}t \\
&= \int_0^1 t^{\alpha - 1} \frac{\partial}{\partial \boldsymbol{\theta}} h(\boldsymbol{\theta}) g'(t^{\alpha} h(\boldsymbol{\theta})) \mathrm{d}t = \left( \int_0^1 t^{\alpha - 1} g'(t^{\alpha} h(\boldsymbol{\theta})) \mathrm{d}t \right) \frac{\partial}{\partial \boldsymbol{\theta}} h(\boldsymbol{\theta}).
\end{aligned}
\tag{13}
$$

Now letting

$$t = \left( \frac{s}{h(\boldsymbol{\theta})} \right)^{1/\alpha},$$

gives

$$\mathrm{d}t = \frac{1}{\alpha h(\boldsymbol{\theta})} \left( \frac{s}{h(\boldsymbol{\theta})} \right)^{(1-\alpha)/\alpha} \mathrm{d}s.$$

It follows that

$$
\begin{aligned}
\mathbf{f}^{\star}(\boldsymbol{\theta}) &= \int_0^{h(\boldsymbol{\theta})} \left( \frac{s}{h(\boldsymbol{\theta})} \right)^{(\alpha - 1)/\alpha} \frac{1}{\alpha h(\boldsymbol{\theta})} \left( \frac{s}{h(\boldsymbol{\theta})} \right)^{(1-\alpha)/\alpha} g'(s) \mathrm{d}s \; \frac{\partial}{\partial \boldsymbol{\theta}} h(\boldsymbol{\theta}) \\
&= \frac{\partial h(\boldsymbol{\theta})/\partial \boldsymbol{\theta}}{\alpha h(\boldsymbol{\theta})} \left( \int_0^{h(\boldsymbol{\theta})} g'(s) \mathrm{d}s \right),
\end{aligned}
$$

and hence,

$$\mathbf{f}^{\star}(\boldsymbol{\theta}) = \left( \frac{g(h(\boldsymbol{\theta})) - g(0)}{\alpha \left( h(\boldsymbol{\theta}) \right)} \right) \frac{\partial}{\partial \boldsymbol{\theta}} h(\boldsymbol{\theta}).$$

If $\boldsymbol{\theta}$ is such that $h(\boldsymbol{\theta}) = 0$, then from (13), we get

$$\mathbf{f}^{\star}(\boldsymbol{\theta}) = \left( \frac{g'(0)}{\alpha} \right) \frac{\partial}{\partial \boldsymbol{\theta}} h(\boldsymbol{\theta}).$$

$\square$

### A.2. Norm Score Approximation for Example 3.4

Consider Example 3.2 with $\phi$ such that such that $c_1 \leq (\phi(t) - \phi(0))^2/t^2 \leq c_2$ for some $0 < c_1 \leq c_2 < \infty$ and for all $t \in \mathcal{T}$ for some set of interest $\mathcal{T}$, e.g., Swish-type or linear output layer. Recall that $\boldsymbol{\theta} = [\boldsymbol{\theta}_1, \ldots, \boldsymbol{\theta}_m]$ where $\boldsymbol{\theta}_j = [a_j, \mathbf{b}_j]$. Also denote $\boldsymbol{\theta}^{\star} = [\boldsymbol{\theta}_1^{\star}, \ldots, \boldsymbol{\theta}_m^{\star}]$ where $\boldsymbol{\theta}_j^{\star} = [a_j^{\star}, \mathbf{b}_j^{\star}]$. We get

$$\|\mathbf{f}^{\star}(\boldsymbol{\theta})\|^2 = \sum_{j=1}^m \|\mathbf{r}^{\star}(\boldsymbol{\theta}_j)\|^2 = \sum_{j=1}^m \gamma_j^2 \left( [\max\left\{ \langle \mathbf{b}_j, \mathbf{x} \rangle, 0 \right\}]^2 + a_j^2 \cdot \|\mathbf{x}\|^2 \cdot \mathbb{1}_{\{\langle \mathbf{b}_j, \mathbf{x} \rangle > 0\}} \right),$$

where

$$\gamma_j \triangleq \frac{\phi(a_j \cdot \max\left\{ \langle \mathbf{b}_j, \mathbf{x} \rangle, 0 \right\}) - \phi(0)}{2 a_j \cdot \max\left\{ \langle \mathbf{b}_j, \mathbf{x} \rangle, 0 \right\}}.$$

For any $\mathbf{x}_i$, denote

$$f_i(\boldsymbol{\theta}) = \sum_{j=1}^{m} r_i(\boldsymbol{\theta}_j) = \sum_{j=1}^{m} \phi_i(a_j \cdot \max\{\langle \mathbf{b}_j, \mathbf{x}_i \rangle, 0\}).$$

Suppose for some $j \in \{1, 2, \ldots, m\}$, $\langle \mathbf{b}_j, \mathbf{x}_i \rangle > 0$, as otherwise $\mathbf{f}_i^\star(\boldsymbol{\theta}) = \mathbf{0}$. We have,

$$\gamma_j^2 a_j^2 \|\mathbf{x}_i\|^2 \leq \sum_{j=1}^{m} \gamma_j^2 \left( [\max\{\langle \mathbf{b}_j, \mathbf{x}_i \rangle, 0\}]^2 + a_j^2 \cdot \|\mathbf{x}_i\|^2 \cdot \mathbb{1}_{\{\langle \mathbf{b}_j, \mathbf{x}_i \rangle > 0\}} \right) \leq \left( \sum_{j=1}^{m} \gamma_j^2 \left( \|\mathbf{b}_j\|^2 + a_j^2 \right) \right) \|\mathbf{x}_i\|^2.$$

By assumption on $\phi$, it follows that

$$c_1 a_j^2 \|\mathbf{x}_i\|^2 \leq \sum_{j=1}^{m} \gamma_j^2 \left( [\max\{\langle \mathbf{b}_j, \mathbf{x}_i \rangle, 0\}]^2 + a_j^2 \cdot \|\mathbf{x}_i\|^2 \cdot \mathbb{1}_{\{\langle \mathbf{b}_j, \mathbf{x}_i \rangle > 0\}} \right) \leq c_2 \left( \sum_{j=1}^{m} \left( \|\mathbf{b}_j\|^2 + a_j^2 \right) \right) \|\mathbf{x}_i\|^2.$$

Assume $a_j^\star \neq 0$ for all $j \in 1, 2, \ldots, m$. Let $l > 0$ be such that $\min_j (a_j^\star)^2 \geq l$ and $0 < u < \infty$ be such that $\sum_{j=1}^{m} \left( \|\mathbf{b}_j^\star\|^2 + (a_j^\star)^2 \right) \leq u$. Define the set

$$\mathcal{C} \triangleq \left\{ [a_1, \mathbf{b}_1, \ldots, a_m, \mathbf{b}_m] \mid \min_{j=1,\ldots,m} (a_j^\star)^2 \geq l, \quad \sum_{j=1}^{m} \left( \|\mathbf{b}_j^\star\|^2 + (a_j^\star)^2 \right) \leq u \right\}.$$

Clearly, by construction, $\boldsymbol{\theta}^\star \in \mathcal{C}$. It follows that that for any $\boldsymbol{\theta} \in \mathcal{C}$, we have

$$c_1 l \|\mathbf{x}_i\|^2 \leq \|\mathbf{f}_i^\star(\boldsymbol{\theta})\|^2 \leq c_2 u \|\mathbf{x}_i\|^2,$$

which in turn gives

$$\min\{c_1 l, 1\} \left( \|\mathbf{x}_i\|^2 + m^2 \phi_i(0)^2 \right) \leq \left\| \widehat{\mathbf{f}}_i^\star(\boldsymbol{\theta}) \right\|^2 \leq \max\{c_2 u, 1\} \left( \|\mathbf{x}_i\|^2 + m^2 \phi_i(0)^2 \right).$$

This implies that

$$\tau_i(\boldsymbol{\theta}) = \frac{\left\| \widehat{\mathbf{f}}_i^\star(\boldsymbol{\theta}) \right\|_2^2}{\left\| \widehat{\mathbf{F}}^\star(\boldsymbol{\theta}) \right\|_{\mathrm{F}}^2} \leq \left( \frac{\max\{c_2 u, 1\}}{\min\{c_1 l, 1\}} \right) \frac{\|\mathbf{x}_i\|^2 + m^2 \phi_i^2(0)}{\|\mathbf{X}\|_{\mathrm{F}}^2 + m^2 \sum_{j=1}^{n} \phi_j^2(0)} = \left( \frac{\max\{c_2 u, 1\}}{\min\{c_1 l, 1\}} \right) \tau_i,$$

where $\tau_i$ is the norm score for the $i^{\text{th}}$ row of

$$\widehat{\mathbf{X}} = \begin{bmatrix} \mathbf{x}_1^{\mathsf{T}} & m\phi_1(0) \\ \mathbf{x}_2^{\mathsf{T}} & m\phi_2(0) \\ \vdots & \vdots \\ \mathbf{x}_n^{\mathsf{T}} & m\phi_n(0) \end{bmatrix} \in \mathbb{R}^{n \times (d+1)}.$$

This allows us to pick $\beta = \min\{c_1 l, 1\} / \max\{c_2 u, 1\}$ in Theorem 3.1.

### A.3. Beyond Nonlinear Least-squares.

Going beyond nonlinear least-squares settings, it turns out that an extension of our adjoint based approach still applies as long as the map $f$ has suitable homogeneity properties. More precisely, consider (1) and suppose $\ell$ is a nonnegative loss and $f$ is positively homogeneous[2] of degree $\alpha$, e.g., ReLU and varaints such as Leaky ReLU. Since $\ell$ is a nonnegative loss function, we define

$$h(\boldsymbol{\theta}) \triangleq \sqrt{\ell(f(\boldsymbol{\theta}))}.$$

---

[2]Recall that a function $\psi$ is positively homogeneous of degree $\alpha$ if $\psi(t\boldsymbol{\theta}) = t^\alpha \psi(\boldsymbol{\theta})$ for any $t > 0$; see (Schechter, 1996) for more details on positive homogeneity.

Analogously to Definition 3.1, for any given $\mathbf{x}$ and $\boldsymbol{\theta}$, we can define

$$
\begin{aligned}
\mathbf{h}^\star(\boldsymbol{\theta}) &\triangleq \int_0^1 \frac{\partial}{\partial\boldsymbol{\theta}} h(t\boldsymbol{\theta}) \mathrm{d}t \\
&= \int_0^1 \frac{1}{2}\left(\frac{\ell'(f(t\boldsymbol{\theta}))}{\sqrt{\ell(f(t\boldsymbol{\theta}))}}\right)\frac{\partial}{\partial\boldsymbol{\theta}} f(t\boldsymbol{\theta})\mathrm{d}t \\
&= \frac{1}{2}\left(\int_0^1 t^{\alpha-1}\left(\frac{\ell'(t^\alpha f(\boldsymbol{\theta}))}{\sqrt{\ell(t^\alpha f(\boldsymbol{\theta}))}}\right)\mathrm{d}t\right)\frac{\partial}{\partial\boldsymbol{\theta}} f(\boldsymbol{\theta},\mathbf{x}),
\end{aligned}
$$

where the last equality follows from Euler's homogeneous function theorem. Now, letting $s = t^\alpha f(\boldsymbol{\theta})$, we have $\mathrm{d}s = \alpha t^{\alpha-1} f(\boldsymbol{\theta})\mathrm{d}t$, which gives

$$
\begin{aligned}
\mathbf{h}^\star(\boldsymbol{\theta}) &= \frac{1}{2\cdot\alpha f(\boldsymbol{\theta})}\left(\int_0^{f(\boldsymbol{\theta})}\frac{\ell'(s)}{\sqrt{\ell(s)}}\mathrm{d}s\right)\frac{\partial}{\partial\boldsymbol{\theta}} f(\boldsymbol{\theta}) \\
&= \left(\frac{\sqrt{\ell(f(\boldsymbol{\theta}))}-\sqrt{\ell(0)}}{\alpha f(\boldsymbol{\theta})}\right)\frac{\partial}{\partial\boldsymbol{\theta}} f(\boldsymbol{\theta}).
\end{aligned}
$$

Since $h(\boldsymbol{\theta}) = h(\mathbf{0}) + \langle \mathbf{h}_i^\star(\boldsymbol{\theta}), \boldsymbol{\theta}\rangle$, we can define

$$
\widehat{\mathbf{h}}^\star(\boldsymbol{\theta}) = \begin{bmatrix}\mathbf{h}^\star(\boldsymbol{\theta}) \\ h(\mathbf{0})\end{bmatrix},
$$

to get

$$
\begin{aligned}
\mathcal{L}(\boldsymbol{\theta}) &= \sum_{i=1}^n \ell(f_i(\boldsymbol{\theta})) = \sum_{i=1}^n (h_i(\boldsymbol{\theta}))^2 \\
&= \sum_{i=1}^n \left(\left\langle\widehat{\mathbf{h}}_i^\star(\boldsymbol{\theta}), \widehat{\boldsymbol{\theta}}\right\rangle\right)^2 = \left\|\widehat{\mathbf{H}}^\star(\boldsymbol{\theta})\widehat{\boldsymbol{\theta}}\right\|^2,
\end{aligned}
$$

where $\widehat{\mathbf{H}}^\star(\boldsymbol{\theta})$ is defined analogously to Definition 3.2. Now, similar to the case of nonlinear least-squares, importance sampling according to leverage scores or norms of $\{\widehat{\mathbf{h}}_i^\star(\boldsymbol{\theta})\}$ give sampling approximations of the form (10).

## A.4. Construction of the $\varepsilon$-Net in Section 3.3.2

Let $\boldsymbol{\theta}^\star$ denote a solution to (1), and consider a compact ball with radius $R$, chosen large enough to contain $\boldsymbol{\theta}^\star$. Let $\mathcal{B}_R^\star$ denote this ball. For any $\varepsilon \in (0,1)$, we pick a discrete subset $\mathcal{N}_\varepsilon \subseteq \mathcal{B}_R^\star$ such that, for every $\boldsymbol{\theta}\in\mathcal{B}_R^\star$, there exists at least one $\boldsymbol{\theta}'\in\mathcal{N}_\varepsilon$ satisfying

$$
\|\boldsymbol{\theta}-\boldsymbol{\theta}'\|_2 \le \varepsilon R.
$$

The construction of an $\varepsilon$-net reduces an uncountable continuous set to a finite covering set; see Figure 3. For completeness, we provide some details here. The reader is encouraged to consult references such as Woodruff et al. (2014); Vershynin (2018) for further discussions and details.

To find an upper bound on the cardinality of the set $|\mathcal{N}_\varepsilon|$, one can use standard volume and covering number arguments. Recall that the volume of $\mathcal{B}_R^\star$ is

$$
\mathrm{Vol}(\mathcal{B}_R^\star) = \frac{\pi^{p/2}R^p}{\Gamma\left(\frac{p}{2}+1\right)},
$$

where $\Gamma$ is Euler's gamma function. A bound on $|\mathcal{N}_\varepsilon|$ can be obtained as the number of small balls with radius $\varepsilon R/2$ that cover the larger ball of radius $(1+\frac{\varepsilon}{2})R$, which is given by the ratio of their respective volumes:

$$
|\mathcal{N}_\varepsilon| \ge \frac{\left(1+\frac{\varepsilon}{2}\right)^P R^p}{\left(\frac{\varepsilon R}{2}\right)^p} = \left(1+\frac{2}{\varepsilon}\right)^p \in \Theta\left(\frac{1}{\varepsilon^p}\right).
$$

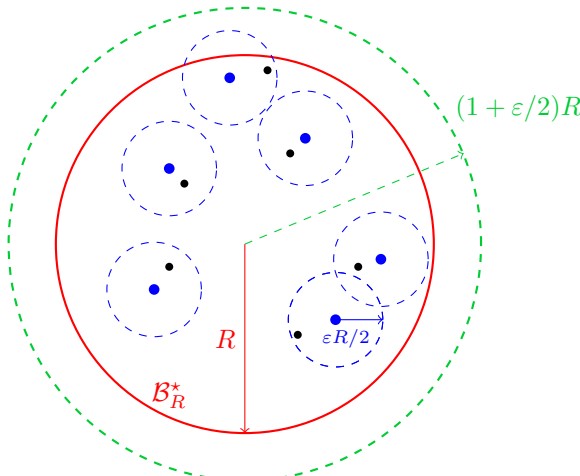

*Figure 3.* Illustration of an $\varepsilon$-net covering $\mathcal{B}_R^\star$. The larger circle has radius $(1 + \varepsilon/2)R$, while $\mathcal{B}_R^\star$ has radius $R$. Blue dots denote the $\varepsilon$-net, and dashed circles of radius $\varepsilon R/2$ cover all points.

## A.5. Further Details for Section 4

**Classification Experiments.** To carry out the experiment in an under-parameterized setting, the dataset was balanced, and the images were resized to $10 \times 10$ dimensions with a grayscale background. A fully connected MLP was trained with a linear 100-input layer connected to a hidden layer with 10 neurons and a ReLU activation unit, followed by a sigmoid output transformation function. The optimal weights were computed using PyTorch, with the Adam optimizer for 1000-5000 epochs (depending on the dataset) and BCEWithLogitsLoss(). These optimal weights were then used to calculate the nonlinear leverage scores in Definition 3.3 for each data point.

**Regression Experiments.** Here, we train a single-index model using a bounded output transformation function described in Example 3.3, with $c_1 = 1$ and $c_2 = 2$. After training for 30,000 epochs, we obtain the optimal parameters and compute the nonlinear importance scores for each data point, as well as the classical linear leverage/row-norm scores using the original data matrix. We then sample training instances using a stratified strategy, proportional to these scores, and evaluate how well each sampling strategy preserves the training performance. Specifically, we solve the subproblem outlined in (12). The experiment is repeated multiple times, and we measure the median log relative error between the MSE of the parameter $\theta_{\mathcal{S}}^\star$ on the full dataset and the optimal MSE obtained by training on the full dataset, i.e., $\log(\mathcal{L}(\theta_{\mathcal{S}}^\star) - \mathcal{L}(\theta^\star))/\mathcal{L}(\theta^\star)$ as a function of the number of samples selected (the number of samples were chosen depending on the total size of the dataset).

## A.6. Quantitative performance for Qualitative datasets

In this section, we show that the datasets presented in Figure 2 serve not only as qualitative illustrations but also as quantitative evidence that sampling based on nonlinear scores helps reduce the training mean squared error (MSE) loss more effectively than alternatives. This reduction is more clearly observed when the loss is plotted on a logarithmic scale. We support this result with experiments on two datasets, SVHN (digits 1 vs. 0) and QD, comparing nonlinear leverage score sampling to linear leverage score sampling and uniform sampling.

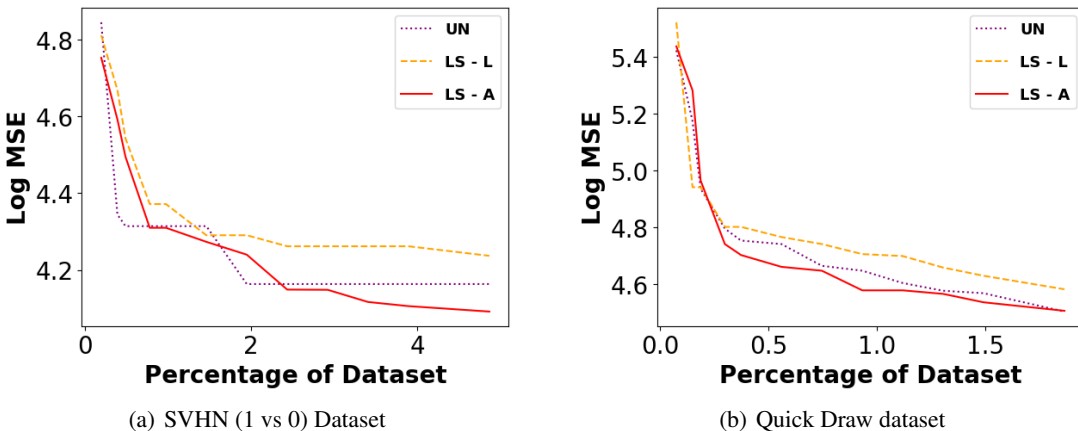

(a) SVHN (1 vs 0) Dataset        (b) Quick Draw dataset

*Figure 4.* Illustration of quantitative results on datasets used in Figure 2, SVHN & QD. The Y-axis shows Log(MSE) on training data against sample size (in percentage of total data). "LS' and "UN" denote Leverage Scores and Uniform Sampling schemas, respectively, with "L" and "A" indicating linear and adjoint-based nonlinear variants.

### A.7. Additional images from Section 4

To facilitate further comparisons, we provide an additional 50 images with the highest and lowest leverage scores for each dataset from the classification experiments in Section 4.

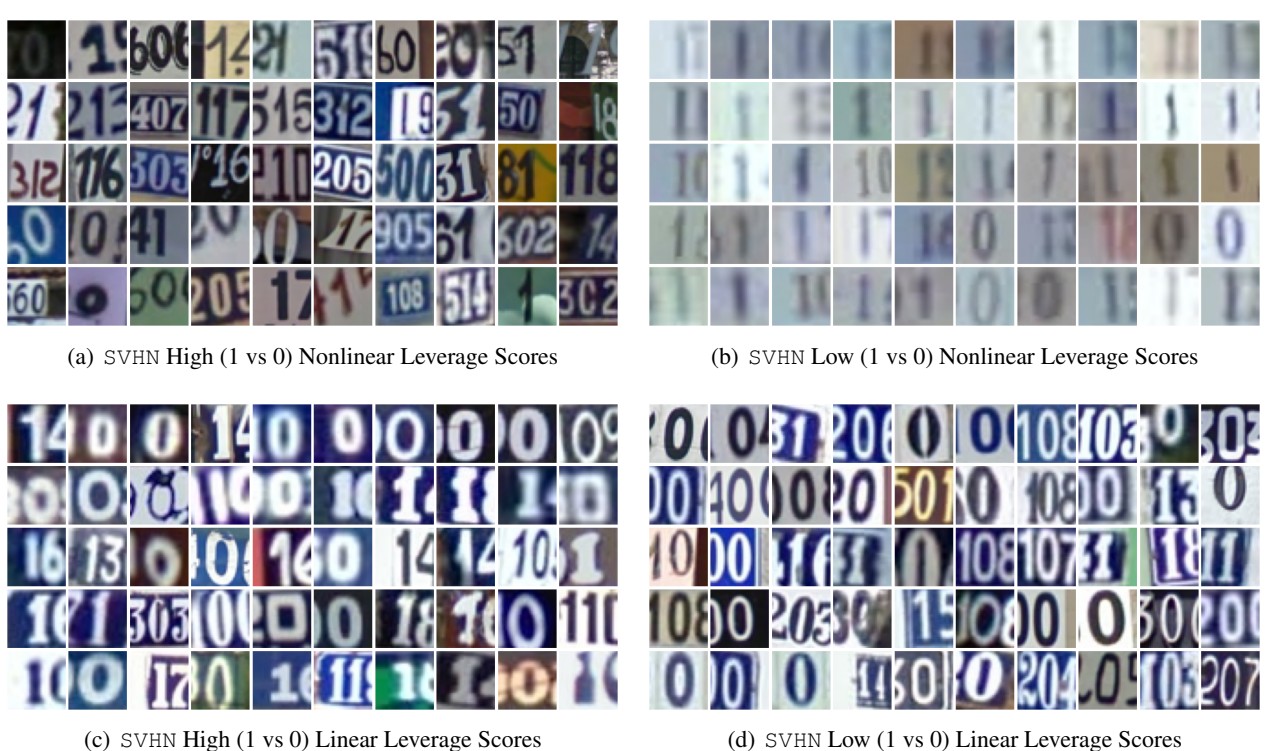

(a) SVHN High (1 vs 0) Nonlinear Leverage Scores      (b) SVHN Low (1 vs 0) Nonlinear Leverage Scores

(c) SVHN High (1 vs 0) Linear Leverage Scores      (d) SVHN Low (1 vs 0) Linear Leverage Scores

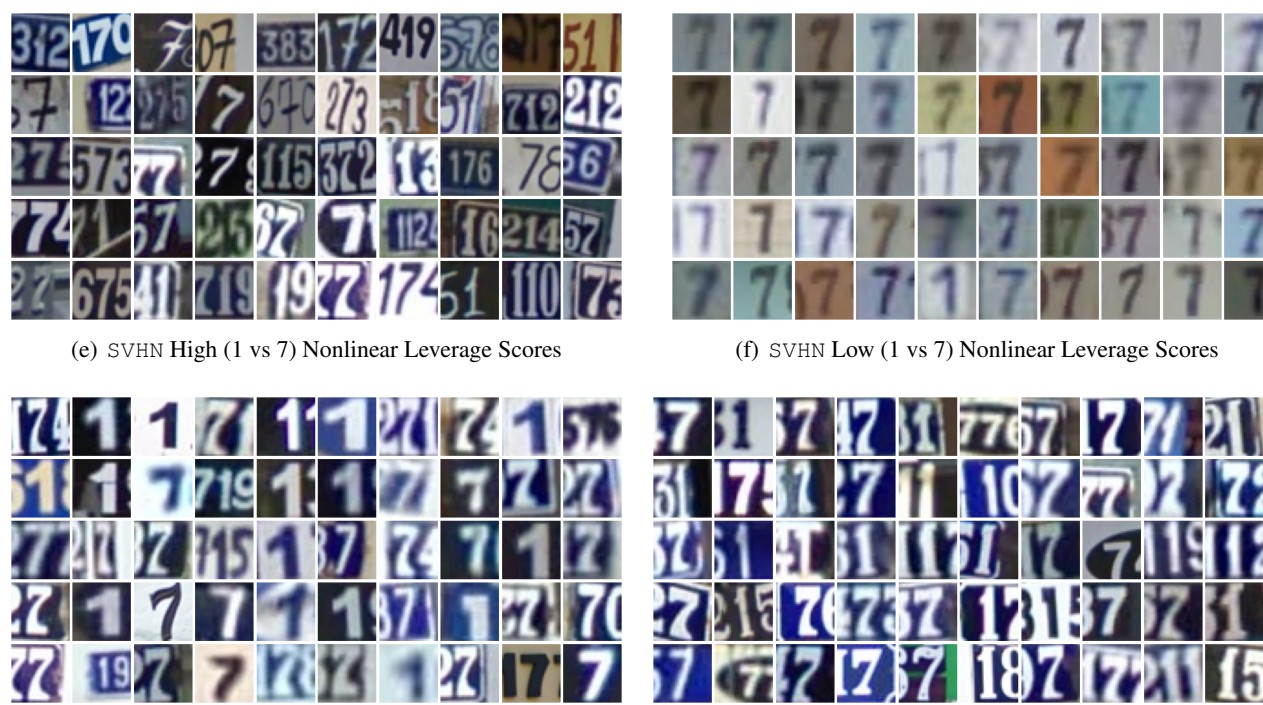

(e) SVHN High (1 vs 7) Nonlinear Leverage Scores    (f) SVHN Low (1 vs 7) Nonlinear Leverage Scores

(g) SVHN High (1 vs 7) Linear Leverage Scores    (h) SVHN Low (1 vs 7) Linear Leverage Scores

Figure 5. Top 50 images with the highest and lowest nonlinear leverage scores in each grouping for the SVHN dataset.

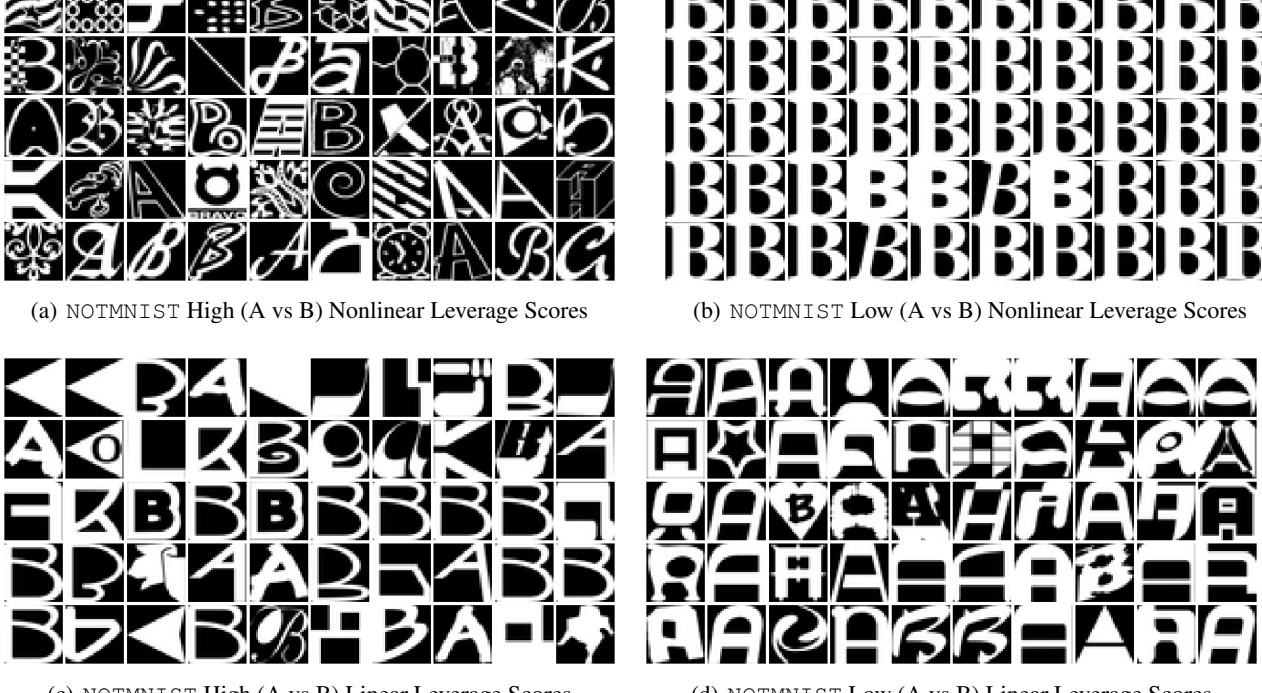

(a) NOTMNIST High (A vs B) Nonlinear Leverage Scores    (b) NOTMNIST Low (A vs B) Nonlinear Leverage Scores

(c) NOTMNIST High (A vs B) Linear Leverage Scores    (d) NOTMNIST Low (A vs B) Linear Leverage Scores

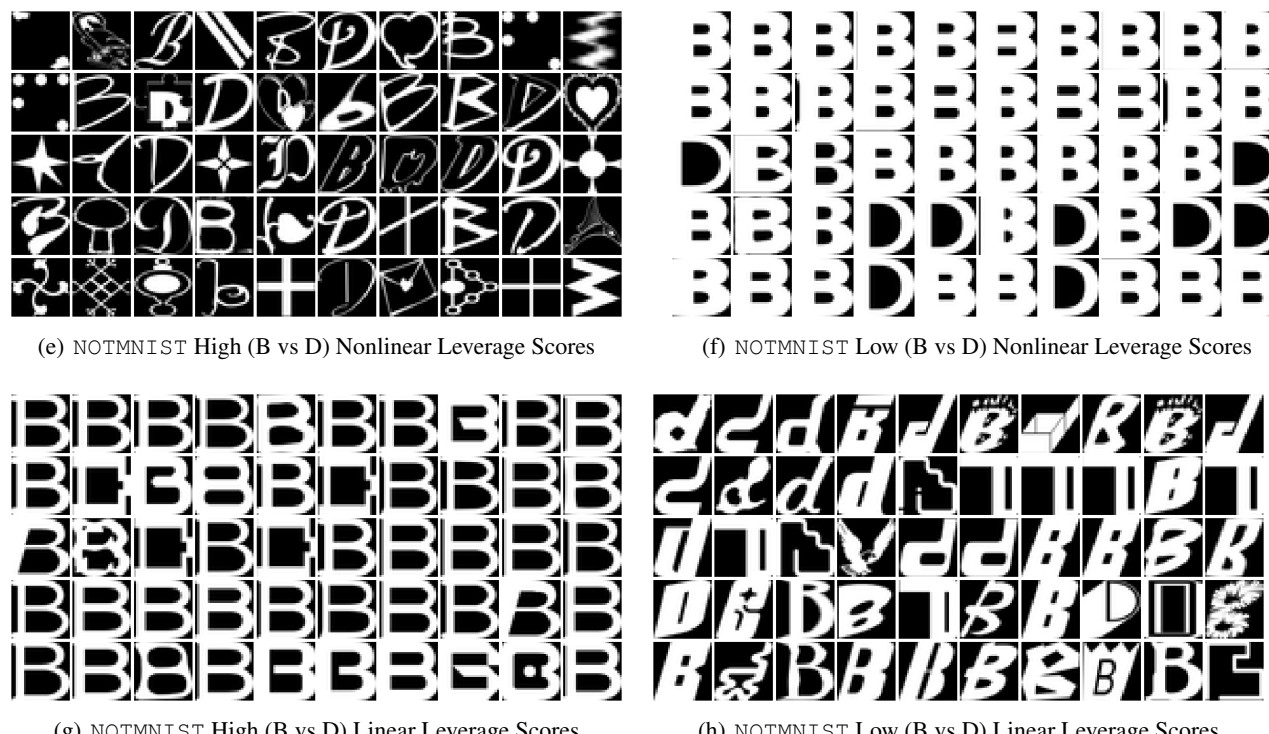

(e) NOTMNIST High (B vs D) Nonlinear Leverage Scores

(f) NOTMNIST Low (B vs D) Nonlinear Leverage Scores

(g) NOTMNIST High (B vs D) Linear Leverage Scores

(h) NOTMNIST Low (B vs D) Linear Leverage Scores

*Figure 6.* Top 50 images with the highest and lowest nonlinear leverage scores in each grouping for the NOTMNIST dataset.

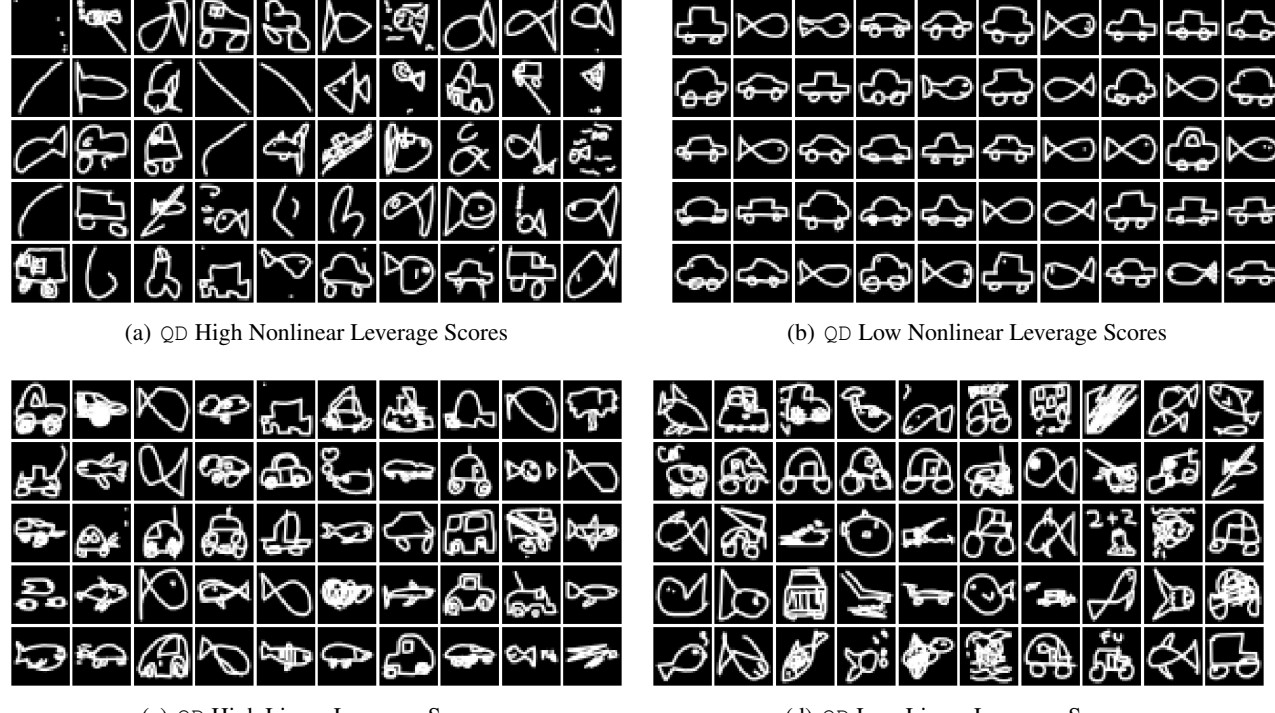

(a) QD High Nonlinear Leverage Scores

(b) QD Low Nonlinear Leverage Scores

(c) QD High Linear Leverage Scores

(d) QD Low Linear Leverage Scores

*Figure 7.* Top 50 images with the highest and lowest nonlinear leverage scores for the QD dataset.

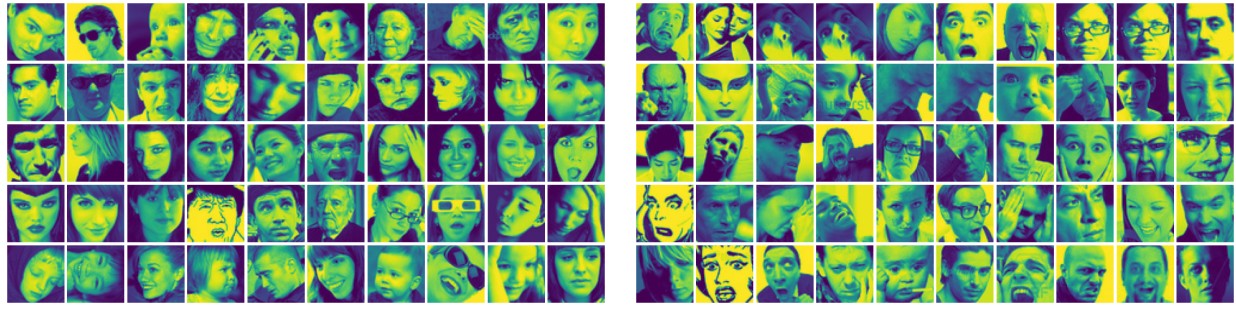

(a) FER - 75% High Nonlinear Leverage Scores

(b) FER - 75% Low Nonlinear Leverage Scores

(c) FER - 65% High Nonlinear Leverage Scores

(d) FER - 65% Low Nonlinear Leverage Scores

(e) FER - 55% Nonlinear Leverage Scores.

(f) FER High Linear Leverage Scores

(g) FER Low Linear Leverage Scores

*Figure 8.* Top 50 (trained and linear settings) and 16 (under-trained setting) images with the highest and lowest nonlinear leverage scores for valence expression on the FER dataset. In (e) only 16 images had non-zero scores at initialization.

