# OpenReview forum: "Importance Sampling for Nonlinear Models"
_ICML.cc/2025/Conference — ICML 2025 poster_

### Official Review · Reviewer_JXa1 · 2025-03-12

**Overall Recommendation:** 3

**Summary:**

The paper introduces a framework to generalize norm-based and leverage-score-based importance sampling from linear models to nonlinear settings. It uses adjoint operator of a nonlinear map, to do so. The authors demonstrate that their sampling methods provide guarantees analogous to linear models, enabling approximation results for nonlinear mappings.

## Update after Rebuttal
I appreciate the response provided by the authors and hope that the next version incorporates the required changes. Hence, I maintain a positive assessment.

**Claims And Evidence:**

The examples and propositions are supported with clear discussions.

**Essential References Not Discussed:**

All major relevant results are cited and discussed in the paper.

**Experimental Designs Or Analyses:**

The experiment section supports the theoretical claims.

- For the classification task, the comparison with L1 Lewis weights (like Mai, Musco, Rao, 2021) and leverage + uniform (like Munteanu, Schwiegelshohn, Sohler, Woodruff, 2018) are missing.

**Methods And Evaluation Criteria:**

Yes it does. It is based on the adjoint operator for non-linear maps, which complements the use of non-linear leverage scores.
The experiments in the paper compare the performance of a model trained on a smaller sample of training data. However, the classification task could be compared with more sampling methods.

**Other Comments Or Suggestions:**

Refered to questions.

**Other Strengths And Weaknesses:**

- $\textbf{Strengths:}$ The paper introduces an innovative way of importance sampling for nonlinear models using the adjoint operator. Further, it offers theoretical guarantees backed by experiments.

- $\textbf{Weakness:}$ The experiments are compared with limited sampling methods. The theorem misses the running time discussion. For data in higher dimensions the quadratic dependence on the coreset size maybe a bottleneck for practical purposes.

**Questions For Authors:**

- Can it be extended to losses other than squared loss?

- Is it possible that the nonlinear leverage scores can be utilized for active learning strategies or transfer learning? If so, then you may also discuss this in the main paper to strengthen it.

- What are the primary computational bottlenecks when directly calculating nonlinear importance scores? How can it be handled for high-dimensional data?

**Relation To Broader Scientific Literature:**

Importance sampling has been extensively studied in linear model problems such as least square regression, clustering, etc. This result is a key towards non-linear subspace embedding which is due to the adjoint operator.

**Theoretical Claims:**

The propositions and examples seemed correct, which was useful for understanding the idea and the practical and relevance.

In the main theorem, the running time analysis is missing.

---

> ### Author Rebuttal · Authors · 2025-03-31
>
> Dear Reviewer JXa1, we sincerely appreciate the time you devoted to reviewing our paper and your comments. We aim to address your comments and questions in detail below.
> ### Additional Sampling Methods in Classification
> Thank you for the suggestion. While adding
> more sampling methods would ideally strengthen our
> arguments, we find that this is unfortunately not feasible, as most prior works are limited in the settings they apply to and do not extend to the nonlinear examples we have considered in the paper. We would like to note that many
> existing methods are more restricted in the types of models they apply to, typically being linear or kernel-based,
> while our approach applies to general nonlinear models via the adjoint perspective. For example, L1 Lewis
> weights have been studied for linear predictor models,
> e.g., logistic regression, but are not trivially extended to
> the nonlinear predictors we consider, e.g., single index
> model. To our knowledge, the only prior works applicable to our setting are that of Gajjar et al. (2023, 2024).
> However, their results rely on linear leverage scores, and
> there is no general construction for nonlinear importance
> sampling like we offer in the paper. In this light, the plots
> in our paper labeled as ”LS - L,” i.e., those based on linear leverage scores, represent the expected performance
> from Gajjar et al. (2023, 2024).
> ### Run Time Analysis
> Our theoretical focus was primarily on sample complexity and its effect on loss approximation, which in turn reduces the time complexity when training a sub-sampled
> dataset. Analyzing runtime depends on various components that are outside the scope of this work, such as the
> choice of optimization algorithm, its hyperparameters,
> and the algorithm used for approximating the nonlinear
> scores. We will add a brief discussion in the revision,
> noting that the cost of approximating these scores is analogous to the linear case and its associated components.
> ### Utilizing Nonlinear Scores for Active/Transfer Learning
> Thank you for raising this important suggestion. While
> we did not explicitly discuss active or transfer learning in
> our paper, the general concept of “nonlinear importance
> scores” can be naturally extend to these areas. In an one-shot active learning setting, for instance, selecting the
> most “informative” points would help reduce labeling
> effort. For transfer learning, one could use these scores
> to identify samples in a source domain that are most
> representative of a target domain’s features. We will
> add a discussion note in the final version outlining these
> possibilities to strengthen the paper.
> ### Quadratic Dependence on $p$
> We appreciate your comment. We recognize that many
> subspace embedding results, including ours, exhibit an
> $\mathcal{O}(p^2)$ sample-size term. However, as an upshot, we
> provide a loss approximation guarantee of the form (2),
> which to our knowledge has not been done prior to our
> work in the context of nonlinear predictor models such as
> neural networks (please see page 3, left column, lines 148–156, as well as
> Remark 3.2 on page 7).
>
> Achieving lower complexity for fully general nonlinear embeddings while maintaining a loss approximation
> guarantee of the form (2) remains an open challenge. In
> our work, we primarily operate in an underparameterized
> regime, where the number of features is not excessively
> large compared to the number of observations, i.e., $n \gg p $, which
> might help mitigate this issue to some extent. While we have noted this limitation in Remark 3.2, we will emphasize it more clearly and reference recent advances in randomized embeddings to highlight potential approaches and future directions for addressing this bottleneck in broader high-dimensional nonlinear settings.
> ### Extension Beyond Squared-loss
> Our Appendix A.3 indicates that the framework extends
> conceptually to other positively homogeneous losses, but
> the complete theoretical guarantees for general losses
> remains for future work.
> ### Primary Computational Bottlenecks For Directly Calculating Nonlinear Importance Scores
> Thanks for your question. For arbitrary nonlinear models,
> computing the adjoint operator may require numerical
> integration if no closed-form expression exists. However,
> in more structured settings, such as single-index models or certain ReLU-based networks, explicit formulas
> described in the main paper allow for standard matrix
> operations to calculate row-norm or leverage scores, similar to the linear case. For row-norm scores, one simply
> evaluates the norm of each row in the “nonlinear dual matrix.” For leverage scores, a QR or SVD-like factorization of the dual matrix is needed, for which standard randomized NLA techniques (e.g., approximate factorizations) can help speed up these factorizations. We will highlight these computational considerations in the next revision of our paper.
> ### Additional Links
> https://anonymous.4open.science/r/ICML2025Review/

---

### Official Review · Reviewer_Z4FF · 2025-03-14

**Overall Recommendation:** 3

**Summary:**

The paper proposed a sampling method for important data by extend the norm and leverage scores in linear models to nonlinear models and reduce the computational complexity.

**Claims And Evidence:**

The proposed methods for important data points in nonlinear models offer several advantages, such as reduced computational complexity, enhanced explainability, and improved outlier detection. These claims are supported by both theoretical analyses and experimental results.

**Essential References Not Discussed:**

N/A

**Experimental Designs Or Analyses:**

see Methods And Evaluation Criteria

**Methods And Evaluation Criteria:**

The paper evaluates the proposed method on four standard datasets, but these datasets are relatively small, and the image sizes are quite tiny. While the results on these smaller datasets are useful for demonstrating the method's functionality, they may not fully represent the challenges of real scenario. The evaluation on larger datasets with higher-dimensional data would provide stronger evidence for the proposed approach.

**Other Comments Or Suggestions:**

see above

**Other Strengths And Weaknesses:**

The paper extends the concept of importance sampling from linear models to nonlinear models, which is highly meaningful, as most neural networks used today are nonlinear. The proposed method can improve the efficiency of training these models.

The paper provides a robust theoretical foundation for the proposed method, which supports its claims effectively.

The experiments demonstrate that the proposed method performs well on small subsets of datasets. However, it would be beneficial to compare the convergence speed of the proposed method, as this would provide additional insights into its practical efficiency.

**Questions For Authors:**

Does the proposed method improve convergence of model training?

How does the proposed method perform on larger datasets?

**Relation To Broader Scientific Literature:**

The paper effectively bridges the gap between existing methods used for linear models to nonlinear models.

**Theoretical Claims:**

The theoretical aspects of the paper are reasonable and well-justified.

---

> ### Author Rebuttal · Authors · 2025-03-31
>
> Dear Reviewer Z4FF, we sincerely appreciate the time you took to review our work. To the best of our ability, we aim to address your comments and questions in detail below.
>
> ### Experiments with High-dimensional Data
> Thank you for your observations and feedback. As mentioned at the outset in the introduction, we make the assumption that $n \geq p$, i.e., the **underparameterized** setting. While the construction of our Adjoint Operator in Definition 3.1 remains
> valid for all dimensions, the nonlinear scores in Definitions 3.3 and 3.4 only provide useful information when
> $n \geq p$. Otherwise, if $p \geq n$, the nonlinear leverage
> scores for all data points would be the same, meaning
> that all data points are considered equally important in
> fitting the model. In this case, our sampling distributions imply a uniform sampling scheme rather than a
> non-uniform importance sampling scheme.
>
> This inherent limitation is also present across the field
> of randomized numerical linear algebra (RandNLA) and
> widely used importance sampling methods such as leverage scores [1]. In fact, for many of the theoretical guarantees in RandNLA to be meaningful, the number of data
> points must often be exponentially larger than the dimension. Unfortunately, extending importance sampling methods like leverage scores in a meaningful way to overparameterized settings, even in the linear case, remains an open problem.
>
> In this light, we would like to clarify that in our setting,
> “high dimensional” is interpreted in terms of the number of observations rather than the number of features.
> While many of the datasets considered in our experiment
> contain large numbers of observations, the underparameterized regime limits us from using more conventional
> datasets that would result in overparameterization and
> large number of features.
>
> [1] Woodruff, D. P. (2014). Sketching as a tool for numerical linear algebra. Foundations and Trends in Theoretical Computer Science, 10(1–2), 1-157.
>
> ### Improving Convergence of Model Training
> Thank you so much for your suggestion. We note that our main focus in the paper and our theoretical results concentrate on
> sampling complexity and its effect on loss approximation
> rather than the convergence speed of a given optimization
> method used for training. As you agree, convergence
> speed can be influenced by many factors, such as the
> choice of the optimization algorithm, selection of hyper-parameters, computing/network architecture, etc. While
> our results are not concerned with these factors, to address your question, we have provided a plot showing
> training time (in seconds) and relative error with respect
> to sample size for subsampling using nonlinear leverage
> score (the results are consistent across different nonlinear sampling methods), offering insight into the tradeoff
> [(please see here)](https://anonymous.4open.science/r/ICML2025Review/Convergence.pdf). As shown, training a model sampled according
> to nonlinear leverage scores achieves $10^{−2}$ relative error
> faster than the time it takes to train the model on the
> entire dataset. This speedup, coupled with the reduction
> in the cost of labeling the additional data, showcases the
> advantages of our nonlinear scores in reducing overall
> costs of training.
>
> ### Additional Links
> 1. New Datasets (Numerical Experimentation): [Please see here](https://anonymous.4open.science/r/ICML2025Review/NewDatasets.pdf)
> 2. Existing Classification Dataset (Numerical Experimentation): [Please see here](https://anonymous.4open.science/r/ICML2025Review/Fig2Quant.pdf)

---

### Official Review · Reviewer_Nx6D · 2025-03-16

**Overall Recommendation:** 3

**Summary:**

This paper introduces a novel family of distributions that extends leverage score distributions—widely used in subset selection for linear models—to nonlinear models. The key component of this construction is a newly defined nonlinear adjoint operator, which satisfies the identity:

$L(\theta) = \|\|\hat{F}^{*}(\theta) \hat{\theta}\|\|^2,$
where $\hat{\theta}:= [\theta|1]$.
This mirrors the classical case in linear regression, where the loss function satisfies:

$L(\theta) = \|\|X \theta - y\|\|^2 = \|\| \hat{X} \hat{\theta}\|\|^2,$
where $\hat{X}:= [X|-y]$.

Leveraging this formulation, the paper explores importance sampling based on these newly introduced nonlinear leverage scores. Specifically, it investigates conditions under which the following bound holds with high probability:

$L(\theta_S) \leq L(\theta^*) + O(\epsilon)$

where $S$ is the index set of $s$ samples drawn via probability weights determined by the nonlinear leverage scores, also referred to as nonlinear norm scores.

The authors focus on two classes of nonlinear models: generalized linear predictors and ReLU neural networks. In both cases, they demonstrate that nonlinear leverage scores can be upper bounded by linear leverage scores. Furthermore, they establish necessary conditions for the high-probability validity of the above bound when using the quadratic loss in a regime where the sample size scales nearly linearly with the dimension.

Finally, numerical simulations illustrate the advantages of sampling based on nonlinear leverage score distributions over traditional linear leverage score distributions.

## Update after rebuttal
I would like to thank the authors for their response. I increase the score from 2 to 3.

**Claims And Evidence:**

- While the claims about the novelty of the construction are valid. The claims about the theoretical guarantees are overstated, since they are only valid for the squared loss, and they are only applicable for a limited family of nonlinear models.

- The claims about the numerical performance are barely substantiated by the shown results: while we can observe that the use of nonlinear leverage scores leads to an improvement compared to the use of the classical leverage scores, the improvement is marginale and barely noticeable (e.g., Figure 1-b). Moreover, for the classification task, the comparison between non linear leverage scores and linear leverage scores is at best qualitative.

**Essential References Not Discussed:**

Existing literature on coresets is missing from the section dedicated to related work (section 2). For instance:
* Determinantal Point Processes for Coresets https://jmlr.org/papers/volume20/18-167/18-167.pdf
* On Coresets for Logistic Regression https://arxiv.org/abs/1805.08571

Overall the comparison with sensitivity sampling schemes is missing in the line of [Langberg and Schulman 2010].

Langberg, M. and Schulman, L.J., 2010, January. Universal $\epsilon$-approximators for integrals. In Proceedings of the twenty-first annual ACM-SIAM symposium on Discrete Algorithms (pp. 598-607). Society for Industrial and Applied Mathematics.

**Experimental Designs Or Analyses:**

See Methods/Evaluation criteria.

**Methods And Evaluation Criteria:**

A quantitative evaluation of the methods was only conducted for two datasets. The rest of the experimental section is based on a qualitative comparison that is not relevant to illustrate the theoretical guarantees established in Section 3.

**Other Comments Or Suggestions:**

-

**Other Strengths And Weaknesses:**

The article is well written, and it is easy to follow for someone who is familiar with the litterature. The construction of the nonlinear adjoint operator is elegant, and the study of this operator would be of interest per se.

A notable weakness is the lack of an extensive empirical validation. The comparison with linear leverage scores was restricted to two datasets, and the observed advantage of the newly introduced scores remains unclear.

**Questions For Authors:**

1) Is there a way to strengthen Theorem 3.1. by assuming that sampling is done according to the $\tau_i(\theta)$ and not $\tau_i$?

2) Could you provide an empirical comparison, similar to Figure 1, for a task other than regression?

**Relation To Broader Scientific Literature:**

The use of importance sampling to build coresets for linear models has been extensively studied in recent years, leading to the development of a new subfield at the intersection of machine learning and randomized linear algebra. While linear leverage scores provide a robust solution to the subsampling problem from both theoretical and empirical perspectives, extending this framework beyond linear models has remained a persistent challenge for the community. Indeed, unlike in the linear case—where the leverage score distribution is independent of the optimal solution to the underlying optimization problem—existing approaches for non-linear models typically involve a chicken-and-egg problem, as the sampling distribution depends on the very parameter being optimized. While the proposed construction of the non-linear adjoint operator is elegant, it does not resolve this fundamental issue.

**Theoretical Claims:**

All proofs were examined, but not carefully checked, and no glaring mistakes were identified.

---

> ### Author Rebuttal · Authors · 2025-03-31
>
> Dear Reviewer Nx6D, we are grateful for the time and effort you devoted to reviewing our paper. We sincerely hope to address your comments and questions in detail below.
> ### Scope of Theoretical Guarantees
> 1. Thank you for your observation. While the examples in the paper (single-index models, ReLU networks) illustrate how the adjoint operator can be computed explicitly from Proposition 3.1, we would like to clarify that the
> method extends conceptually to broader classes of functions via its natural definition (Definition 3) through
> numerical integration.
> 2. While our approach can conceptually extend to broader
> nonlinear models, we agree that our current theoretical
> guarantees focus on squared losses. Appendix A.3 outlines initial steps toward more general losses, but fully
> developing the theory in this setting remains an open
> challenge we are pursuing. We will make sure to clearly
> state these limitations in the revised version.
> ### Numerical Experimentation (Figure 1)
> 1. We agree that incorporating additional datasets would enhance the quality of our empirical evaluations. To address this, we have now included two additional datasets–one for classification & one for regression [(Please see here)](https://anonymous.4open.science/r/ICML2025Review/NewDatasets.pdf).
> 2. We believe our method can offer substantial performance gains in many cases. The perception otherwise
> may stem from our choice of axis scales, which make the graphs appear less visually impressive. In our numerical experiments, we used a log-scale on the Y-axis. However, we should have better highlighted that even a small shift on
> a log-scale can represent a significant absolute difference. For example, in the California Housing Prices dataset,
> comparing linear row-norm with non-linear row-norm, the shift from −0.75 to −2 in log-scale corresponds to a
> relative error reduction from 18% to 1%. We will clarify this more clearly in the text.
> ### Qualitative Interpretation of Classification Tasks (Figure 2)
> We agree that our current results for the classification task rely on qualitative interpretation rather
> than quantitative analysis. Our main goal was to
> demonstrate the perceptual interpretability and diagnostic power of our framework, highlighting that our nonlinear scores capture meaningful information about the data
> that would otherwise be unavailable. To our knowledge,
> this is the first approach of its kind, introducing a new
> interpretability & diagnostics paradigm for classification
> with nonlinear models. However, we fully agree that incorporating quantitative
> metrics would strengthen our work. To address this, we
> have added two additional figures that compare the quantitative performance on two existing datasets, namely
> SVHN & QD [(Please see here)](https://anonymous.4open.science/r/ICML2025Review/Fig2Quant.pdf).
> ### Fundamental Challenges
> Our theoretical guarantees are flexible in that the underlying sampling distribution only requires approximations to the nonlinear scores that are independent of the parameter being optimized. In many cases, as demonstrated in Examples 3.3 and 3.4, such parameter-independent estimates can be obtained, making our approach less susceptible to the “chicken-and-egg” problem you mentioned.
>
> However, obtaining a solution $\theta_{S}^{\star}$ that satisfies (2) requires solving a constrained optimization problem, where the constraint set must be large enough to contain the true solution $\theta^{\star}$. Hence, while our approach relaxes the dependency of the sampling distribution on the parameters being optimized, it still relies on a constraint that implicitly assumes some prior knowledge of $\theta^{\star}$. As a result, the core issue you highlighted remains an inherent challenge.
>
> Addressing this “chicken-and-egg” problem in broader nonlinear contexts remains an open problem. We will ensure that these limitations are discussed more prominently.
> ### Literature on Coresets
> Thank you for pointing this out, and we apologize for
> the oversight. Although we briefly mentioned the idea
> of coresets and their applicability beyond linear embeddings on page 2, we will ensure that the final version
> includes a dedicated related work section on coreset
> frameworks.
> ### Strengthening Theorem 3.1
> Thank you for the thoughtful question. Our approach
> aims to construct a sampling distribution that is independent of the parameter being optimized. Accordingly,
> our theory allows for an approximation of $\tau_i(\theta)$ in the
> form of $\tau_i$, which serves as a uniform bound for all $\theta$. As you noted, this can lead to conservative guarantees, potentially requiring a larger sample size than necessary.
> In our experiments (Figure 2 (k, i)), we observe that importance scores become more informative as the model nears optimality. Motivated by this, we are exploring how to replace the uniform bound with one at the optimal point, $\tau_i(\theta^{\star})$, and its implications for theory. A full development is left for future work.

---

### Decision · Program_Chairs · 2025-05-01

**Decision:**

Accept (poster)

**Comment:**

The paper uses the notion of adjoint operator and generalizes norm-based and leverage-score-based methods from linear to non-linear settings. The paper provides some theoretical analysis and considers the setting of generalized linear and RELU neural networks. It has limited experimental evaluations. I am leaning towards an accept if there is room in the program.